# WHY TRANSFORMERS SUCCEED AND FAIL AT COMPOSITIONAL GENERALIZATION: COMPOSITION EQUIVALENCE AND MODULE COVERAGE

## ABSTRACT

Compositional generalization—the ability to train on some combinations of modules and then generalize to unseen module combinations—is an important form of out-of-distribution generalization. A large body of work has evaluated this form of reasoning in transformer-based models, but the underlying mechanisms of success and failure remain poorly understood. We systematically evaluate compositional generalization in transformer-based models, and we identify two factors that play important roles in determining performance: ***composition equivalence*** and ***module coverage***. We show that the apparent performance of direct models (trained only on final outputs) can be entirely due to exploiting composition equivalences—different sequences of modules that reduce to identical end-to-end functions. When benchmarks eliminate these equivalences, the performance of these models drops to *near zero*, showing their inability to generalize to compositions of known modules that produce novel end-to-end functions. We discuss two key failure modes of step-by-step learning (trained on intermediate outputs). We show that composition equivalences encourage shortcut learning in step-by-step models, and these models fail to generalize when specific modules always appear at certain positions or in fixed combinations in the training set. These findings provide new insights into the conditions under which atomic modules that constitute a compositional task can be correctly learned by a model class for a specific train-test distribution.

## 1 INTRODUCTION

Many real-world tasks require reasoning about novel combinations of familiar components. Examples include reasoning about the output of a new software program built from known modules, planning novel sequences of actions to accomplish a given task, or constructing a novel proof from known logical operations. As transformer-based models are increasingly used in real-world applications, such as software development, robotics, and scientific discovery, it is useful to know the conditions under which they will reliably exhibit compositional reasoning.

In this work, we focus on understanding the extent to which transformer models exhibit *task-based compositional generalization*—the ability to generalize to unseen module combinations after being trained on a limited set of such combinations. We study task-based compositional generalization in the context of one of its simplest forms: sequential compositional tasks. Such tasks consist of a sequence of "atomic" modules that transform input data into output data (see Figure 1(a) for an example). One effective way for a model to achieve sequential compositional generalization is for it to learn internal representations that implement the behavior of these atomic modules, allowing it to generalize to arbitrary compositions of these modules. However, many open questions remain about the extent to which transformer-based models can correctly identify and learn the behavior of atomic modules when trained on data with compositional structure.

Substantial theoretical and empirical work has focused on understanding the compositional generalization abilities of transformers (Hupkes et al., 2020; Csordás et al., 2021; Ontanon et al., 2022; Wang et al., 2024; Song et al., 2025; Ahuja & Mansouri, 2025; Lippl & Stachenfeld, 2025). Theoretical work suggests that specific transformer architectures can identify and learn atomic modules

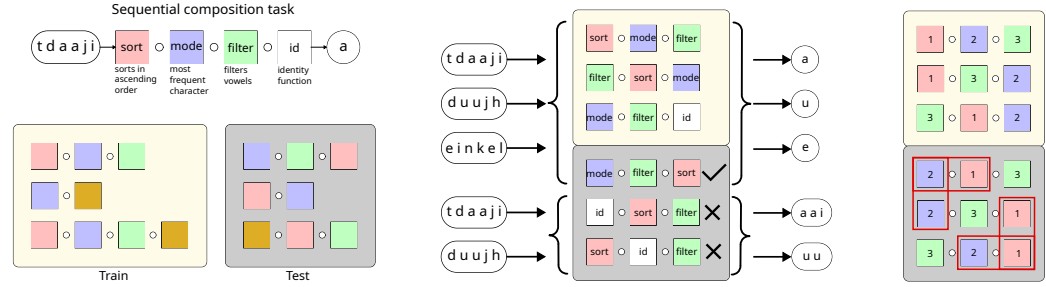

(a) Task-based compositional generalization  (b) Composition equivalence  (c) Module coverage

Figure 1: (a) A sequential composition task combines "atomic" functions (e.g., sort, mode, filter). **Task-based** compositional generalization is the ability to generalize to unseen sequences after training on a subset. (b) **Composition equivalence:** Different sequences are composition equivalent if they reduce to the same end-to-end function. For example, one equivalence class consisting of the first four sequences reduces to returning the most frequently occurring vowel in a set of strings; direct models succeed on the test sequence (mode ○ filter ○ sort) from this class but fail on the remaining two test sequences from a different composition equivalence class. (c) **Module coverage:** Examples of coverage failures include *position-wise* (e.g., function 2 never appearing in the first position) and *pair-wise* (e.g., missing the ordered pair (2,1) in the training set.)

and can exhibit strong compositional generalization and other forms of out-of-distribution generalization (Zhou et al., 2024; Ramesh et al., 2024; Ahuja & Mansouri, 2025; Lippl & Stachenfeld, 2025; Abedsoltan et al., 2025). However, empirical evaluation has shown that these models often fail to generalize over a large variety of unseen module combinations, creating a gap between theoretical claims and empirical performance. Various hypotheses have been proposed to explain this discrepancy between theory and practice, including shortcut learning (Dziri et al., 2023), having an insufficient number of forward passes available (Ramesh et al., 2024), lacking explicit training on autoregressive compositional structure (Abedsoltan et al., 2025), and insufficient model capacity (Peng et al., 2024). However, these explanations have primarily focused on *model characteristics* to explain successes and failures in compositional generalization.

In this work, we focus on the *properties of the data-generating process* that can significantly impact the compositional generalization abilities of transformer models. Through systematic experimentation,[1] we identify two types of distribution shifts between the train and test sets that explain the significant variability observed in the performance of transformer-based models: ***composition equivalence*** and ***module coverage***. *Composition equivalence* occurs when distinct sequences of atomic modules in the train and test data reduce to identical end-to-end functions. *Module coverage* is defined as the extent to which each atomic module is observed at different positions and in similar contexts between train and test data.

Through the concepts of composition equivalence and module coverage, we systematically study compositional generalization across transformer variants: direct models (trained on final outputs) and step-by-step models (trained on intermediate outputs). Our key findings can be summarized as follows:

- The compositional generalization performance of transformer models varies significantly between *within-k settings* (in which train and test compositions consist of the same number of modules) and *cross-k settings* (in which train and test compositions consist of different numbers of modules).

- Transformers can learn *equivalences* at the *composition level*—learning that different sequential compositions perform identical end-to-end mappings.

- Direct models *appear* to compositionally generalize when train and test splits share *composition equivalences*—module sequences that are equivalent in their overall end-to-end function, but not in their composition structure. However, their performance drops to *near zero* when these composition equivalences are eliminated.

---

[1]Code is available at: https://github.com/anonymous-submission-cs/task_based_cg

- Failures of module coverage create *spurious correlations*, negatively impacting the compositional generalization performance of both direct and step-by-step models.

## 1.1 RELATED WORK

In this section, we explain how our work fills important gaps in the understanding of the conditions under which compositional generalization can occur in transformers. An additional related work discussion is provided in A.2.

**Task-based compositional generalization in transformers:** Recent work on task-based compositional generalization (Ramesh et al., 2024; Abedsoltan et al., 2025) has shown that step-by-step models can generalize to an exponential number of sequences, but direct models (trained directly on final outputs) often fail to generalize compositionally. Our findings on the existence of composition equivalences (or the lack thereof) in training data explain the lack of generalization observed in direct models. Previous work has also observed that direct model performance improves when the types of module functions in the composition change (Ramesh et al., 2024; Xu et al., 2024). Our findings on composition equivalence show how different module functions produce different degrees of composition equivalence that can be exploited by direct models. The sensitivity of the models to the selection of module orderings (i.e., module coverage) has also been studied in previous work (Ramesh et al., 2024; Abedsoltan et al., 2025). Our work demonstrates that module coverage failures can negatively impact the learning of composition equivalences in direct models.

**Relation to functional equivalence and coverage criteria:** Recently, Chang et al. (2025) discussed functional equivalence and coverage principles in compositional generalization—seemingly related concepts to composition equivalence and module coverage. However, our work differs significantly. Chang et al. focus on *data-based* compositional generalization, which has also been studied by others (Dziri et al., 2023; Ahuja & Mansouri, 2025). Data-based compositional generalization is defined as the ability to generalize to new data combinations within a fixed task, such as multiplication. In contrast, we study *task-based* compositional generalization. The key distinction is that the labeling function is fixed in data-based generalization but dependent on the composition of module functions in task-based generalization, requiring extrapolation to unseen labeling functions, which makes it more challenging. As a result, functional equivalence defined by Chang et al. (2025) operates at the input data level, i.e., two inputs are functionally equivalent if they return the same output under a *fixed* function. In contrast, composition equivalence operates at a higher abstraction level, defining similarity between compositions that share the same end-to-end function.

## 2 SYSTEMATIC EVALUATION OF TASK-BASED COMPOSITIONAL GENERALIZATION

We first formally define the setup of task-based compositional generalization and then present the evaluation results across different train-test distributions and models.

### 2.1 TASK-BASED COMPOSITIONAL GENERALIZATION

We adapt the formalism proposed by Ramesh et al. (2024), and use it to describe the train-test distribution shifts that we study. Consider a set of $n$ module functions $\mathcal{F} = \{f_1, f_2, \ldots, f_n\}$ where each function $f_i : \mathcal{V}^m \to \mathcal{V}^m$ maps input sequences to output sequences of equal length $m$ over vocabulary $\mathcal{V}$. Assume the input sequence is denoted as $\mathbf{x} = (x_1, \ldots, x_m) \in \mathcal{V}^m$ and the output sequence is denoted as $\mathbf{y} = (y_1, \ldots, y_m) \in \mathcal{V}^m$. A sequential composition of length $k$ is defined as the composition of $k$ functions applied to inputs to obtain outputs, formally expressed as $\mathbf{y} = (f_{i_k} \circ f_{i_{k-1}} \ldots \circ f_{i_1})(\mathbf{x})$, where $f_{i_j} \in \mathcal{F}$. Intermediate outputs of $j$ compositions are denoted as $\mathbf{y}_j$, where $j \in \{1, 2, \ldots k\}$.

In sequence-to-sequence models, sequential composition can be represented by concatenating task tokens and input data tokens such that, the input sequence is $s = (t_1, t_2, \ldots, t_k, x_1, x_2, \ldots, x_m)$, where $t_j \in \mathcal{T}$ are task tokens corresponding to module functions $\mathcal{F}$ and $x_i \in \mathcal{V}$ are data tokens. The task space over sequences of length $k$ is defined as $\mathcal{T}^k$, representing all possible sequences of $k$ task tokens from the task token vocabulary $\mathcal{T}$. Let $\mathbf{T} = (T_1, \ldots, T_k)$, $\mathbf{X} = (X_1, \ldots, X_m)$,

$\mathbf{Y} = (Y_1, \ldots, Y_m)$ denote the vector of random variables corresponding to task, data, and output sequences.

**Direct models:** Direct models are trained autoregressively on a data set consisting of final outputs $\mathcal{D}_{direct} := \{(\mathbf{t}_i, \mathbf{x}_i, \mathbf{y}_i)\}_{i=1}^N$, and learns a complex mapping from input sequences to final output sequences $h_{direct} : \mathcal{T}^k \times \mathcal{V}^m \rightarrow \mathcal{V}^m$, $h_{direct} \in \mathcal{H}_{\text{direct}}$.

**Step-by-step models:** Step-by-step models trained on data with intermediate and the final outputs $\mathcal{D}_{sbs} := \{(\mathbf{t}_i, \mathbf{x}_i, \mathbf{y}_{(1:k)_i})\}_{i=1}^N$. The model learns a mapping from input sequences to intermediate and final output sequences $h_{\text{sbs}} : \mathcal{T}^k \times \mathcal{V}^m \rightarrow \mathcal{V}^{km}$, $h_{\text{sbs}} \in \mathcal{H}_{\text{sbs}}$.

**Train and test distributions:** Let $\mathcal{P}_{\text{train}}$ and $\mathcal{P}_{\text{test}}$ be the train and test distributions over input sequences. The full support for the joint task–data space is $\mathcal{S} = \mathcal{T}^* \times \mathcal{V}^m$, where $\mathcal{T}^* = \bigcup_{k=1}^{k_{\max}} \mathcal{T}^k$. We assume that data tokens are sampled uniformly from $\mathcal{V}^m$ in both train and test distributions, i.e., $\mathbf{X} \sim \text{Uniform}(\mathcal{V}^m)$. We mainly focus on compositional generalization over tasks where train and test distributions have mutually exclusive support over task sequences, i.e., $\text{supp}(\mathcal{P}_{\text{train}}^\mathcal{T}) \cap \text{supp}(\mathcal{P}_{\text{test}}^\mathcal{T}) = \emptyset$.

**Task-based compositional generalization** is defined as the ability of models to generalize to unseen sequences in $\mathcal{P}_{\text{test}}$, when trained on sequences from $\mathcal{P}_{\text{train}}$.

The key research question is: *Under what conditions do the direct and step-by-step models accurately predict outputs for test sequences sampled from $\mathcal{P}_{test}$ when trained on sequences from $\mathcal{P}_{train}$?*

## 2.2 EVALUATION SETTINGS

We present a systematic evaluation of the compositional generalization capabilities of direct and step-by-step models over a wide variety of systematically constructed train and test sets, as described below.

**Uniform vs. diverse set of module functions:** We evaluate compositional generalization on two benchmarks by varying the set of available module functions. The first benchmark is similar to that proposed by Ramesh et al. (2024), consisting of six random bijection functions that all belong to the same function class, which we call the *uniform* benchmark. Each bijection maps an input character to an output character based on a pre-defined lookup table.

Since real-world compositional reasoning tasks might consist of modules of varied complexity and might not be exactly random in their logic, we also create a *diverse* benchmark, inspired by string manipulation functions in software programs and RASP primitives that transformer models can represent and learn (Weiss et al., 2021; Zhou et al., 2024). The module functions are: {sort, concatenate, filter, shift, union, mode}. The logic of these functions is: shift shifts each character by one position to the right in the alphabet (e.g., a $\rightarrow$ b, z $\rightarrow$ a); sort rearranges characters in lexicographic order; mode returns the most frequent character (lexicographically smallest in case of a tie); concatenate appends the second string to the first; union returns ordered unique characters preserving first occurrence order; and filter extracts vowels while maintaining their original order. Module definitions are provided in Appendix A.3.2.

**Within-$k$ and cross-$k$ generalization:** We examine two kinds of task-based generalization depending on the sequence length. In *within-$k$ generalization*, both the training and test distributions are samples from disjoint subsets of the fixed-length task space $\mathcal{T}^k$, corresponding to all possible permutations of the same combination of $k$ functions, where $k \in \{2, \ldots 6\}$. We randomly sample 80% of the permutations and evaluate on the remaining 20% of the permutations.

In *cross-$k$ generalization*, training data includes samples from the composition of $k$ modules $\mathcal{T}^k$ but evaluated on the composition of $k'$ modules $\mathcal{T}^{k'}$, where $(k \neq k')$. Cross-$k$ evaluation of compositional generalization allows us to evaluate to what extent the models can generalize to complex compositions from simpler ones and vice-versa. For implementation purposes, to allow evaluation of sequences with different lengths, we use a dummy identity function, as explained below.

**Including identity module as a dummy module function:** To enable cross-$k$ generalization with fixed input length, we introduce an identity module token $t_{\text{id}} \in \mathcal{T}$ corresponding to $f_{\text{id}}(\mathbf{x}) = \mathbf{x}$. For sequences of length $k < k_{\max}$, we uniformly distribute $(k_{\max} - k)$ identity tokens across module positions to avoid out-of-distribution prompts at test time. For example, if $k = 2$, and $k_{\max} = 7$, for a given module sequence (mode, sort), one of the padded module sequences is: (id, id, mode,

`id, id, sort, id`). For fair comparison between within-$k$ and cross-$k$, we also evaluate within-$k$ settings both with and without identity modules to assess sensitivity to identity modules. Ramesh et al. (2024) merged within-$k$ and cross-$k$ using identity as a dummy module, but we separate them, as we demonstrate in our results that performance differs *significantly* between these settings.

**Models:** We train four variants of autoregressive transformer models—direct and step-by-step models with absolute and relative positional embeddings. We use the nanoGPT architecture (Karpathy, 2023) with three layers and six attention heads (see Appendix A.4 for more details). We implement both absolute and relative positional encoding schemes (Shaw et al., 2018), as relative positional embeddings have demonstrated effectiveness for length generalization (Kazemnejad et al., 2023). We see that it is also effective for compositional generalization, especially cross-$k$ generalization. Training data includes $100k$ samples, and test data includes $10k$ samples, where the number of samples per unique sequence is uniformly distributed. Input data tokens are of length six, randomly sampled from the vocabulary consisting of lowercase alphabets $\mathcal{V} = \{a, b, c, \dots z\}$. More details about data format and training can be found in Appendix A.5.

## 2.3 LARGE VARIABILITY IN COMPOSITIONAL GENERALIZATION PERFORMANCE

Experimental results are shown in Figures 2 and 3. We report the mean accuracy on unseen sequences using exact match scoring (1.0 for perfect predictions and 0.0 otherwise). Performance variability is computed by training the model five times with different random seeds. Our experiments demonstrate substantial variation in compositional generalization performance across both direct and step-by-step models with different positional encodings and train/test distributions. The key findings can be summarized as:

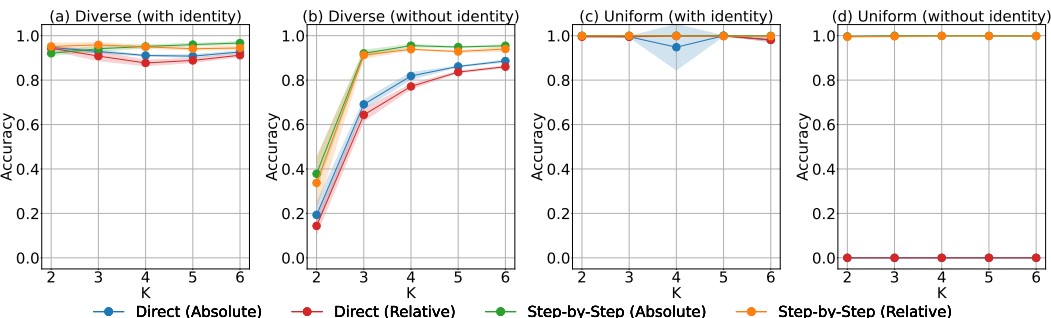

Figure 2: **Within-$k$ evaluation:** Direct models fail completely (0% accuracy) on uniform bijections without identity tokens, while step-by-step models maintain near-perfect performance across all conditions. We observe that including identity tokens or different function types seems crucial for the generalization performance of direct models.

(1) **Direct models fail on the *uniform* (random bijection) benchmark without identity modules, while maintaining reasonable performance on the diverse benchmark:** Figure 2 presents within-$k$ evaluation results. Comparing Figure 2(a) and (b) for the diverse benchmark, the performance of direct models drops from roughly 95% to 80% when identity tokens are excluded. However, for the uniform benchmark consisting of random bijections (Figure 2(c) vs. (d)), the performance of direct models drops from 100% to ***near zero***. Step-by-step models maintain near-perfect performance across all settings, except $k = 2$ in Figure 2(b). Strong performance of direct models on the diverse benchmark or with identity modules, but *near zero* performance on the uniform benchmark without identity modules, implies that module functions play an essential role.

(2) **Significant performance differences between within-$k$ and cross-$k$ evaluation. Cross-$k$ performance is highest for the evaluation $k'$ closer to training $k$.** Figure 3 shows cross-$k$ results, where train-$k$ = 6 is in (a) and (c), and train-$k$ = 3 is in (b) and (d), and evaluation is done on all possible permutations (including identity tokens) across all $k \in \{1, 2, \dots, 6\}$. We observe that step-by-step models with relative position embeddings achieve the best overall performance across all $k$ values. We also observe that when training on $k = 3$, step-by-step models appear to be learning the behavior of atomic functions, as they exhibit near-perfect performance for $K = 1, 2, 4,$ and $5$. How-

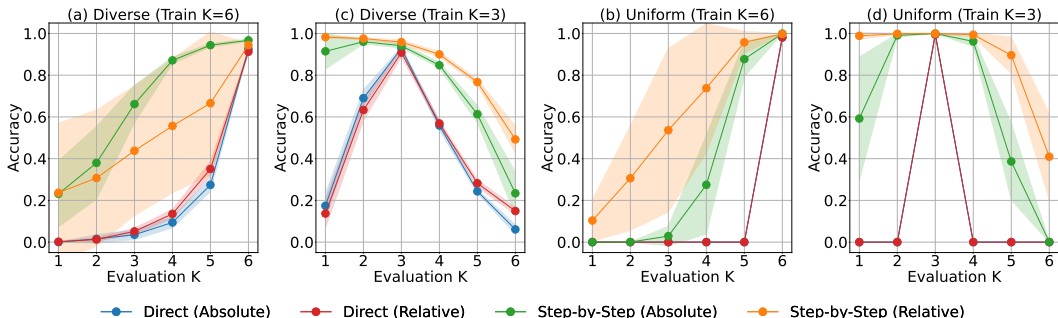

Figure 3: **Cross-$k$ evaluation:** Overall, step-by-step models with relative position embeddings achieve the best cross-$k$ compositional generalization. For the diverse benchmark, direct models show the best performance for evaluation k values closer to training k. Direct models fail on uniform functions even with identity tokens, showing that identity tokens only help within-k evaluation.

ever, when we see the results for train $k = 6$, this behavior is no longer visible, with low performance scores across all datasets and position embeddings. We also observe significant differences between absolute and relative positional embeddings for step-by-step models.

This shows that the compositional generalization performance of these models is not robust for all values of train-$k$, as the module distribution in cross-$k$ changes significantly due to the distribution of the identity modules.

**(3) Direct models perform worse for cross-$k$ generalization and identity modules only help in within-$k$ generalization.** Direct models maintain 40%-60% accuracy for test $k$ values closest to training $k$ with diverse functions, but their performance again drops to zero for the uniform benchmark. Importantly, while identity tokens helped achieve perfect performance for direct models in within-$k$ evaluation with uniform functions (Figure 2(c)), they fail to improve cross-$k$ performance (Figure 3(c,d)), indicating that identity tokens only benefit within-$k$ generalization. We show additional results for the remaining combination sizes in the Section A.6.

On analysis of failure examples of direct and step-by-step models, we identify the interplay between two key train-test distribution shifts that explain the successes and failures of these models: (1) *composition equivalence* based shift, in which train and test sets have sequences that reduce to exactly or approximately identical end- to-end functions, and (2) *module coverage* based shift, in which train and test vary in terms of whether modules appear at the same positions or in the same relative context. We formalize these in the next two sections.

## 3 EXPLAINING COMPOSITIONAL GENERALIZATION PERFORMANCE VIA COMPOSITION EQUIVALENCE

Two *distinct* sequential compositions are defined as equivalent in terms of final outputs if they reduce to the exact input-output mapping for a given set of data tokens. More formally,

**Composition equivalence:** Let $F = f_1 \circ f_2 \circ \cdots \circ f_k$ and $F' = f'_1 \circ f'_2 \circ \cdots \circ f'_k$ be two sequential compositions. We say that $F$ and $F'$ are **equivalent** over an input subspace $\mathbf{X} \in \mathcal{S} \subseteq \mathcal{V}^m$ if they produce identical input-output mapping according to the end-to-end labeling function $g : \mathcal{V}^m \to \mathcal{V}^m$ for the final output $\mathbf{Y} \in \mathcal{V}^m$:

$$f_1 \circ f_2 \circ \cdots \circ f_k(\mathbf{x}) = g(\mathbf{x}) = \mathbf{y} \text{ and } f'_1 \circ f'_2 \circ \cdots \circ f'_k(\mathbf{x}') = g(\mathbf{x}') = \mathbf{y}' \quad \forall \mathbf{x}, \mathbf{x}' \in \mathcal{S}$$

**Composition equivalence class:** A composition equivalence class is a set of sequential compositions that all reduce to the same input-output labeling function $g$ for all inputs sampled from subspace $\mathcal{S}$: $[F]^g_{\mathcal{S}} = \{F' : F' \text{ is equivalent to } F \text{ w.r.t. } g \text{ over } \mathcal{S}\}$

**Identity-based equivalence class:** Let $\mathrm{id} : \mathcal{V}^m \to \mathcal{V}^m$ denote the identity function. For any sequential compositions $F = f_1 \circ f_2 \circ \cdots \circ f_k$, the following set of $k$ sequences belongs to the same equivalence class: $F_1 := f_1 \circ f_2 \circ \cdots \circ f_k \circ \mathrm{id}, F_2 := f_1 \circ f_2 \circ \cdots \circ \mathrm{id} \circ f_k, \ldots, F_k := \mathrm{id} \circ f_1 \circ f_2 \circ \cdots \circ f_k$.

**Identity-based composition equivalence causes near-perfect performance of direct models in within-$k$ evaluation:** For within-$k$ evaluation with an identity module, *random* sampling of the sequences creates train-test splits consisting of composition equivalences, resulting in almost-perfect direct model performance. However, when we remove identity modules, the performance drops for the diverse benchmark (still non-zero) and becomes exactly zero for the uniform benchmark.

Identity-based equivalence represents an *exact* form of composition equivalence over the full input space $\mathcal{V}$. In the uniform benchmark consisting of random map functions (bijections), this is the only form of equivalence possible, since different permutations or combinations of non-identity functions induce distinct end-to-end labeling functions. As a result, direct models exhibit **zero** compositional generalization in two cases: (1) within-$k$ evaluation without identity modules (Figure 2(d)), and (2) cross-$k$ evaluation regardless of identity modules (Figure 3(c),3(d)). Cross-$k$ failures occur because identity-based composition equivalence exists only among sequence lengths with fixed $k$.

Thus, benchmarks consisting of random module functions where each unique permutation of non-identity functions reduces to novel input-output labeling functions are the most challenging, and direct models fail entirely in those cases. Note that (Ramesh et al., 2024) used this benchmark but included identity modules interleaving between module functions, which might explain the success of direct models in some of their experiments.

**Exact and approximate equivalence in the *diverse* benchmark causes significant performance in direct models:** In the diverse benchmark, we observed that the inclusion of module functions such as {`mode`, `filter`} introduces equivalences for a set of input strings due to the invariance of these functions to some input characters. More specifically, `filter` extracts vowels, `mode` selects the maximum-occurring character. Including these functions enables shortcut reasoning, allowing the final answer to be arrived at in some cases without needing to reason accurately through each step. Similarly, `concatenate` and `union` are similar in logic and give the same answer for a pair of strings, if the strings consist of distinct characters. The composition of `shift` (if shifting preserves lexicographic order) and `sort` is commutative, i.e., `shift(sort(x))` = `sort(shift(x))` for a wide variety of strings. An example of approximate equivalence is provided in the Figure 1(b).

Direct models exploit the exact and approximate composition equivalence due to the above function properties to *superficially* achieve significant compositional generalization. But these models fail on sequences that define novel end-to-end functions. This explains their strong performance across all combinations in within-$k$ evaluation (Figure 2(b)) and for nearby training $k$ values in cross-$k$ generalization (Figure 3(a), (b)). Equivalences may also arise across different combination sizes when a function does not affect the overall labeling function (e.g., `sort` after `mode`).

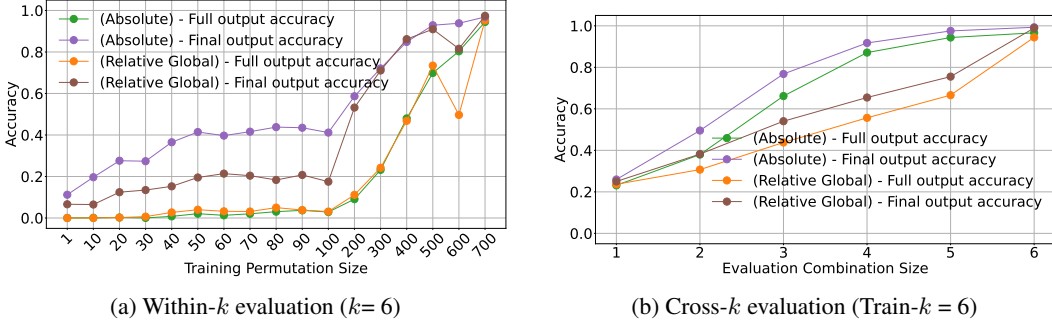

(a) Within-$k$ evaluation ($k$= 6)          (b) Cross-$k$ evaluation (Train-$k$ = 6)

Figure 4: **Composition equivalence encourages shortcuts in step-by-step models:** We observe that composition equivalences in the diverse benchmark cause discrepancies between final output accuracy and full output (intermediate and final output) accuracy for both within-k evaluation and cross-k evaluation. Final accuracy > full accuracy implies the model is learning shortcuts to reach the final answer. If we eliminate composition equivalences, the performance difference is zero between full and final outputs (see Figures 12(a), 12(b))

**Composition equivalences encourage learning of shortcuts in step-by-step models:** Training on the intermediate outputs breaks composition equivalences based on end-to-end labeling functions, improving the identifiability of individual labeling functions. Still, equivalences can also arise at the intermediate output level.

We find that such equivalences encourage step-by-step models to rely on shortcuts rather than step-by-step reasoning, creating multiple paths to the correct answer. In the diverse benchmark consisting of a large number of equivalences, this results in higher final output accuracy but lower step-by-step accuracy, as models often produce incorrect intermediate outputs while still providing the correct final result (Figures 4(a),(b)). In contrast, for the uniform benchmark, where no equivalences exist, step-by-step and final output accuracy exactly match as shown in Figures 12(a), 12(b). Thus, composition equivalence can *negatively* impact compositional generalization in step-by-step models by promoting shortcuts over accurate step-by-step reasoning.

**On necessity and sufficiency of composition equivalence:** We evaluate whether all non-zero direct model performance stems *only* from composition equivalence, and the number of sequences per equivalence class required in the training data to enable the learning of corresponding test equivalences.

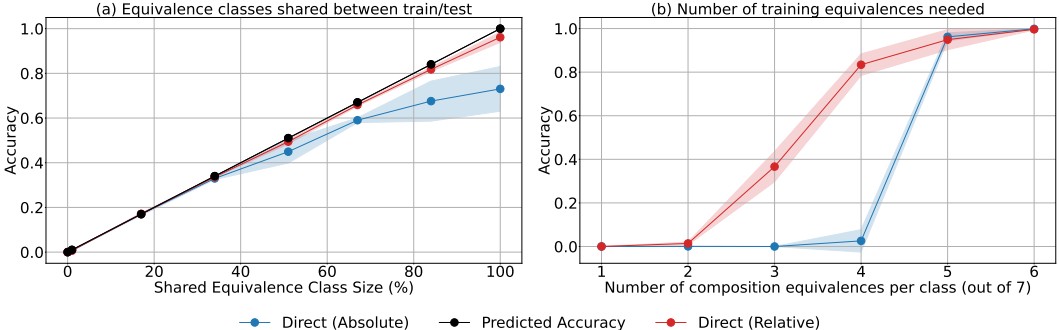

Figure 5: **Equivalence class necessity and training requirements:** (a) Generalization performance to unseen sequences is correlated strongly with the number of shared composition equivalence classes between train and test. (b) Models need to see at least 4–5 composition equivalences (per class) in training to generalize to the corresponding test equivalences.

**Number of shared equivalence classes strongly predicts performance of direct models:** Figure 5(a) shows that direct model performance, for both absolute and relative embeddings, is strongly predicted by the number of shared equivalence classes. We systematically increase the number of shared identity-based equivalent classes in the train and test sets. We use the uniform benchmark because there is a one-to-one mapping between sequences and classes, making it simpler to control the number of shared equivalences. However, the mere existence of shared classes does not always guarantee correct generalization; for example, the absolute positional embedding model saturates in performance after 51% shared classes in Figure 5(a). Upon further analysis, we find that most failures correspond to test sequences that have the identity module in the first position (Figure 25). This highlights the importance of module coverage, which we explore in the next section.

**Four-six training composition equivalences per class required to generalize to corresponding equivalences in the test set:** We vary the number of equivalent training sequences per class from 1 to 6 in (Figure 5(b)). We observe that models require at least four to six equivalents per class for accurate generalization, with absolute position encoding more demanding than relative. The requirement for absolute embeddings is greater than that of relative positional embeddings.

**Understanding the effect of composition equivalence for the diverse benchmark:** In the diverse benchmark, equivalence classes are not pre-defined as in the case of the uniform benchmark. We use the following approach to create splits with varying degrees of composition equivalences by learning equivalence classes from data.

- Compute the pairwise composition equivalence score between two task sequences as the proportion of inputs for which the final outputs match when evaluated over a large number of inputs sampled from the vocabulary.

- Construct an undirected graph based on the equivalence scores, where a node denotes a composition sequence and an edge denotes the strength of the score based on a threshold. As a conservative measure, we set a threshold of 0.01 to create disjoint splits.

- Learn equivalence classes by finding connected components in the graph.

- **Disjoint splits:** Split train and test data by equivalence class—no class appears in both sets. We consider 50-50 splits to accommodate varying degrees of equivalence, as explained below.

- **Splits with varying degrees of composition equivalences:** We increase shared equivalences by swapping roughly half of the members of each test equivalence class with members from a training class. This systematically increases test sequences with shared equivalences in proportion to class sizes, while maintaining a constant test set size. We also tried just leaking test members to training without swapping, which gradually reduced the test set size as the percentage of shared equivalences increased, and observed similar results (see Appendix A.8.2).

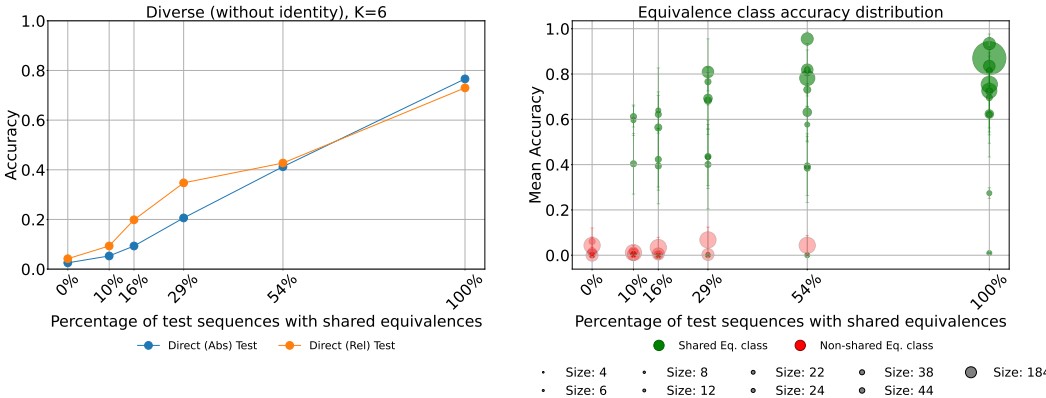

Figure 6: **Isolating the effect of composition equivalence in the diverse benchmark: (K = 6):** (a) Generalization performance to unseen sequences increases with the number of shared composition equivalence classes between train and test. (b) The direct model generalizes mainly to test sequences with shared equivalence classes and performs poorly on classes with novel end-to-end behavior.

**Generalization performance increases with the number of shared equivalences:** In Figure 6(a-b), we observe that accuracy increases sharply from 0% (disjoint) to roughly 80% (100% shared). Unlike results in Figure 5(a), the model doesn't reach perfect accuracy even with 100% leakage because equivalences can be approximate within classes, and some classes contain only 4-6 members, making half the members insufficient to guarantee perfect performance on the unseen half. This shows that the existence of shared equivalence classes during training is necessary for *significant* compositional generalization in direct models, but it is insufficient.

**Direct models primarily generalize to compositions in shared equivalence classes:** Figure 6(b) shows the accuracy distribution per equivalence class from the splits considered in Figure 6(a). We observe that as the number of shared equivalences increases, the direct model's accuracy is significantly higher for shared than for non-shared classes. Figure 15 provides examples of success/failure cases of sequences belonging to two equivalence classes and demonstrates that failure cases primarily belong to non-shared equivalences. The significant performance gap between 0% and 100% shared equivalence splits in both uniform and diverse benchmarks (Figures 5, 6) implies that composition equivalence is a primary mechanism driving *superficial* composition generalization in direct models.

## 4 THE ROLE OF MODULE COVERAGE IN COMPOSITIONAL GENERALIZATION

In this section, we first define a metric that quantifies the degree to which module coverage shifts between the training and test sets, and then evaluate how compositional generalization performance is affected by this shift.

We define a metric that captures divergence in the *position-wise* and *pair-wise* distributions of modules between the test and training sets. Position-wise coverage shift evaluates the extent to which models learn spurious correlations between absolute positions and module functions due to oversampling at specific positions. Pairwise coverage captures whether certain combinations consistently co-occur, causing models to learn them as composite modules rather than compositionally.

For position $i \in \{1, 2, 3, \ldots, k\}$ and module index $j \in \{1, 2, 3, \ldots, m\}$, **position-wise coverage distribution** can be empirically calculated as: $\hat{P}(T_i = t_j) = \frac{1}{N} \sum_{n=1}^{N} \mathbb{I}(T_i^{(n)} = t_j)$, where $T_i^{(n)}$ denotes the module at position $i$ in sequence $n$, averaged across all $N$ sequences in the dataset. The distribution $\hat{P}(T_i)$ is an $n$-dimensional probability vector representing the probabilities of each of the module tokens $t_j$ appearing at a given position $i \in \{1, 2, \ldots, k\}$, with probabilities summing to 1. The empirical **pairwise adjacency distribution** can be calculated as: $\hat{P}(T = t_j, T' = t'_j) = \frac{1}{N(k-1)} \sum_{n=1}^{N} \sum_{i=1}^{k-1} \mathbb{I}(T_i^{(n)} = t_j \text{ and } T_{i+1}^{(n)} = t'_j)$. We consider the KL divergence to quantify module coverage shift between splits, as it is asymmetric, and our goal is to capture shifts in the test set with respect to the training set. Position-wise KL divergence between train and test sets can be defined as $D_{\text{pos}} = \frac{1}{k} \sum_{i=1}^{k} D_{KL}(P_{\text{test}}(T_i) \| P_{\text{train}}(T_i))$ and $D_{\text{pair}} = D_{KL}(P_{\text{test}}(T, T') \| P_{\text{train}}(T, T'))$. We use a linear combination of both divergences $D_{\text{combined}} = 0.5 \cdot D_{\text{pos}} + 0.5 \cdot D_{\text{pair}}$.

Our experiments evaluate the effects of module coverage shift by using two sampling regimes: (1) *random sampling* selects compositions *uniformly* from a fixed-$k$-size sequence space $\mathcal{T}^k$ such that module functions are uniformly distributed across module positions, (2) *systematic sampling serially* selects compositions in order, e.g., $(f_1, f_2, f_3, f_4, f_5, f_6)$, $(f_1, f_2, f_3, f_4, f_6, f_5)$, $(f_1, f_2, f_3, f_6, f_5, f_4)$, ... $(f_6, f_5, f_4, f_3, f_2, f_1)$. We sample 100–700 training compositions in increments of 100. Systematic sampling creates a lack of overlap in the train-test module coverage. For example, for $n = 600$, $f_6$ ***never*** appears in the first position in the training set.

In Figure 7, we can see that uniform sampling creates low KL divergence $(< 0.12)$ between train and test sets, leading to almost perfect accuracy for all models, while systematic sampling creates an extensive range of divergence $(0.8 - 5)$ with perfect performance at low values, zero performance at high values and significant variability in between showing lack of reliability of models in these out-of-distrbution scenarios.

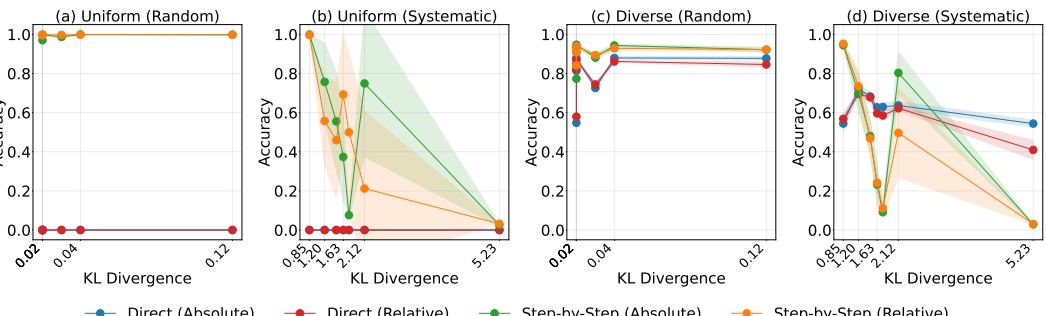

Figure 7: **Random vs. systematic selection of compositions:** (a,c) Uniform sampling creates low values of coverage divergence $(< 0.12)$, leading to almost perfect accuracy for all models (b,d) Systematic sampling creates an extensive range of divergence $(0.8 - 5)$ with perfect performance at low values, zero performance at high values and significant variability in between.

## 5 CONCLUSION

We introduce the novel concept of composition equivalence as a key mechanism through which direct models appear to achieve compositional generalization. We show that direct models often extrapolate by learning composition equivalent sequences rather than learning atomic modules or decomposing individual sequences. This highlights a critical challenge in benchmark design: Should we eliminate equivalences to truly assess compositional reasoning, or include them to reflect real-world conditions where semantic similarities can be exploited? Our findings emphasize the importance of analyzing compositional generalization through the lens of the data-generating process and identifiability, showing how composition equivalences and module coverage failures can lead to shortcut learning in both direct and step-by-step models. While our experiments used synthetic benchmarks with GPT-2-style transformers, extending this analysis to real-world benchmarks and large-scale pre-trained models is a promising future direction.

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

## A APPENDIX

### A.1 LARGE-LANGUAGE MODEL USAGE

We have used large language models to aid and polish writing minimally in the main paper and to a reasonable extent in the Appendix.

### A.2 RELATED WORK

In this section, we discuss additional related work.

**Other studies of compositional generalization in transformers** Beyond the work discussed in 1.1, several other works consider compositional generalization within transformer models, investigating the phenomenon in different settings. Several work (Petty et al., 2024; Csordás et al., 2021;

Ontanon et al., 2022; Zhang et al., 2024) investigate the effect of different hyperparameters and architectural choices on the performance of transformers on various generalization tasks. Others (Li et al., 2023a; Garg et al., 2022; Xu et al., 2024; An et al., 2023) have considered the compositional generalization capabilities of transformers when trained on samples using in-context learning. Several works (Li et al., 2023b; Wei et al., 2022; Zhou et al., 2023; Li et al., 2024) investigate compositional generalization in the context of chain-of-thought prompts, finding gains in performance in various compositional tasks. Yang et al. (2024) evaluates the performance of compositional capabilities of large-language models after instruction-tuning by testing on unseen combinations of instructions. Another line of work investigates the internal circuitry that is learned to enable compositional generalization (Song et al., 2025; Wang et al., 2024), finding specific parts of the transformer architecture that affect generalization capability. Our work contributes to this area of research by investigating the effect of module orderings and data-generating characteristics on the compositional generalization of transformer models.

**Training bias and shortcut learning:** Shortcut learning (Geirhos et al., 2020; Du et al., 2023) is the phenomenon where models rely on superficial features in the training data, which may have spurious correlations with the output, instead of the robust features that capture the true underlying data-generating process. We can view shortcut learning as one of the consequences of the composition equivalence and module coverage violations. Understanding whether data contains composition equivalence or violates module coverage provides a principled approach to mitigate spurious correlations and understand the cases where the model would exhibit robust out-of-distribution generalization.

## A.3 Module Function Definitions

### A.3.1 Uniform Benchmark

We define six random bijection functions $f : \mathcal{V}^m \to \mathcal{V}^m$ that randomly map each input data token $x_i \in \mathcal{V} := \{a, b \ldots z\}$ to a random output data token $y_i \in \mathcal{V} := \{a, b \ldots z\}$, based on a predefined lookup table. We ensure there is a one-to-one, unique mapping between input and output data tokens, and each input data token doesn't map to itself. We assume input data length $m = 6$. The six tokens are sampled uniformly at random from $\mathcal{V}$ without replacement.

### A.3.2 Diverse Benchmark

1. `shift(x)`: The shift function applies a predetermined *bijective* transformation to each character in the input string according to a fixed character substitution table.
   `shift(h j f s d h) = i k g t e i`

2. `sort(x)`: The sorting function rearranges the characters of the input string in lexicographic (ascending alphabetical) order.
   `sort(c g m a h b) = a b c g h m`

3. `mode(x)`: The mode function returns the character that appears most frequently in the input string. In case of equal frequencies, the lexicographically smallest character is selected.
   `mode(w j d n k k) = k`

4. `concatenate(x, y)`: The concatenation function performs string concatenation, appending the second string to the end of the first string.
   `join(s e w l r z, y s e o q n) = s e w l r z y s e o q n`

5. `union(x, y)`: The union function returns the ordered union of unique characters from both input strings, preserving the order of first occurrence within each string.
   `union(s e w l r z, y s e o q n) = s e w l r z y o q n`

6. `filter(x)`: The filter function extracts all vowel characters from the input string while preserving their original order.
   `filter(d c f o j i) = o i`

7. `identity(x)`: The identity function returns the input string unchanged.
   `identity(x) = x`

We sample two input data tokens of length six from the vocabulary $\mathcal{V} = \{a, b, c, \ldots z\}$, as some of the module functions are binary. The maximum possible length of output is 12. We ignore the remaining data tokens if the length exceeds 12, to maintain a fixed output length. We pad with the space token in case output is less than length 12, for example, in case of `mode`, `filter`, `union` functions.

## A.4 TRANSFORMER ARCHITECTURE

We use a modified `nanoGPT` implementation Karpathy (2023) with three transformer layers, each containing a multi-head causal self-attention block (6 heads, embedding size 120), layer normalization, and an MLP with GELU activation. The model has a context window of 36 tokens for direct models and 114 tokens for step-by-step models. The vocabulary size is around 40 tokens. We disable dropout and biases in LayerNorm layers, and apply weight tying between the token embedding and the output projection layer. Both absolute and relative (global) positional encodings were tested in separate experiments.

Models are trained with an autoregressive objective, predicting the next token given the previous sequence. For a sequence $x_{1:T}$ of length $T$, the training loss is the cross-entropy:

$$L(w) = -\sum_{t=1}^{T-1} \log p_w(y = x_{t+1} \mid x_{1:t}),$$

where $p_w$ denotes the model distribution parameterized by weights $w$.

We train for 100 epochs with a batch size of 512. The optimizer is AdamW with $\beta_1 = 0.9$, $\beta_2 = 0.95$, and weight decay 0.1. The learning rate follows a cosine annealing schedule with warmup (100 steps), starting from $3 \times 10^{-4}$ and decaying to $6 \times 10^{-6}$. Gradient clipping is applied with a maximum norm of 1. Training is performed on a single GPU using PyTorch 2.0, with Flash Attention kernels (`scaled_dot_product_attention`) enabled when available.

### A.4.1 ABSOLUTE AND RELATIVE POSITIONAL EMBEDDINGS

We experiment with two positional encoding embeddings: *absolute* embeddings and *relative global* embeddings.

**Absolute embeddings.** In the absolute case, we follow the original transformer formulation (Vaswani et al., 2017), where a learnable position embedding $P(t)$ is added to the token embedding $E(x_t)$ before being passed into the transformer layers:

$$z_t = E(x_t) + P(t), \quad t = 1, \ldots, T.$$

This ensures that positional information is directly encoded in the input sequence representation, with the embedding table learned jointly with the model. In our implementation, the transformer instantiates both token and position embedding tables, which are added elementwise at each time step.

**Relative global embeddings.** In the relative case, we replace the absolute embedding table with a learned relative representation incorporated into the attention mechanism. This approach is motivated by the relative position representations of Shaw et al. (2018) and extended in the Music Transformer (Anna et al., 2018). Formally, each attention head maintains a trainable relative embedding matrix $E_r \in \mathbb{R}^{C \times d_h}$, where $C$ is the maximum context length and $d_h$ the head dimension. Given query vectors $Q \in \mathbb{R}^{B \times H \times T \times d_h}$, we compute relative logits as:

$$S_{\text{rel}} = \text{skew}(Q E_r^\top),$$

where the skew operation aligns relative positions with their corresponding offsets. These logits are then added to the standard dot-product attention $QK^\top / \sqrt{d_h}$, yielding attention weights that encode both content and relative distance. This modification removes the absolute embedding table, instead parameterizing $E_r$ within each attention block.

## A.5 TRAINING DETAILS

**Data generation.** We generate $100K$ training samples evenly distributed across the unique train permutations and $10k$ test samples evenly distributed across the unique test permutations. For example, for $k = 6$, there are a total 720 module permutations without identity module–so we sample roughly 138 samples per permutation with random data strings combined in train and test sets. All input data tokens are of fixed-size strings of length six and sampled uniformly without replacement from the $\mathcal{V} = \{a - z\}$. For binary functions in the diverse benchmark, we sample two input strings of length 6, and the output can be of max-length 12, from the same vocabulary space. As the uniform benchmark consists of unary bijection functions, both input and output data tokens are of length 6.

**Distinct input data strings:** As we randomly sample input strings of length six without replacement, there are a total of 11,576,560 unique strings. In our datasets, we found approximately 99.9% unique strings across both the training and test sets of total size 110k (100k + 10k = 110k). .

**Prompt format:** To generate the training sequences for the transformers, we serialize a vocabulary consisting of lowercase alphabet characters along with special tokens <START>, <SEP>, <END>, and a space token. <START> and <END> mark the sequence boundaries, while <SEP> separates function tokens, input strings, intermediate outputs (for step-by-step data), and the final output. Spaces are used for padding to ensure fixed input lengths across examples.

We evaluate module orderings on two sets of functions. In the *uniform* case, all functions are bijective map operations and require only a single input string. In the *diverse* case, where functions are based on common string operations, functions may take up to two input strings and therefore each prompt must contain two input sequences. With this in mind, the following is the exact prompt structure used to train and evaluate our models:

For data points generated with the *direct* setting, the prompt structure is:

$$\texttt{<START>}\ f_1 f_2 \ldots f_k\ \texttt{<SEP>}\ x_1 \ldots x_n\ [\texttt{<SEP>}\ x'_1 \ldots x'_n]\ \texttt{<SEP>}\ y_1 \ldots y_m\ \texttt{<END>},$$

where $f_1, \ldots, f_k$ denote the module composition, $x_1 \ldots x_n$ the first input string, $[\texttt{<SEP>}\ x'_1 \ldots x'_n]$ the second input string present only in the diverse setting, and $y_1 \ldots y_m$ the final output.

In the *step-by-step* setting, the intermediate outputs of each function are also included between separators, yielding multiple <SEP> segments:

$$\texttt{<START>}\ f_1 f_2 \ldots f_k\ \texttt{<SEP>}\ x_1 \ldots x_n\ [\texttt{<SEP>}\ x'_1 \ldots x'_n]\ \texttt{<SEP>}\ x^{(1)}\ \texttt{<SEP>}\ x^{(2)} \ldots \texttt{<SEP>}\ y\ \texttt{<END>},$$

where $x^{(i)}$ denotes the intermediate output after applying the $i$th function in the composition.

**Positional embeddings and length generalization** Positional embeddings have been a major area of research when considering the length generalization of transformer architectures. Length generalization in transformers is the behavior where models trained on shorter sequences still perform well on longer ones unseen in the training data. In a sense, this requires that the transformer generalize seen modules at one position to perform the same task at another. This framing provided some of the motivation for including relative embeddings in transformers–the idea is that sequences of set behaviors should be learned relative to each other instead of at an absolute position. Indeed, several works (Neishi & Yoshinaga, 2019; Ruoss et al., 2023) have shown that using relative positional embeddings outperforms absolute ones. We refer readers to the survey by Zhao et al. (2024) for a larger collection of community efforts. It is this same motivation that informs our decisions to consider relative positional embeddings. Though we do not test length generalization (since our input prompts are fixed length), we instead consider scenarios where modules do not appear in specific positions or appear less frequently.

## A.6 ADDITIONAL CROSS-$k$ COMPOSITIONAL GENERALIZATION RESULTS

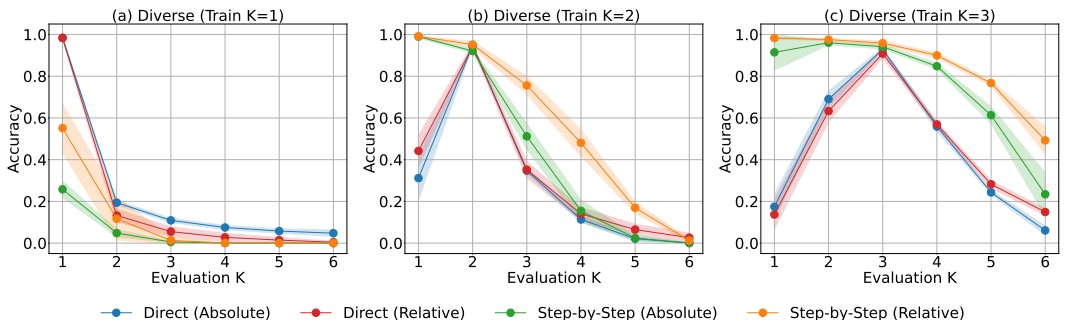

Figure 8: **Cross-k evaluation (Diverse), train-$k$ = 1, 2 and 3**

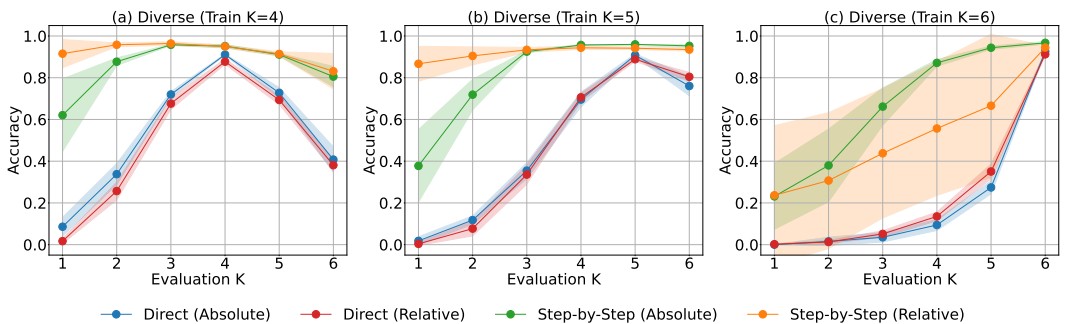

Figure 9: **Cross-k evaluation (Diverse), train-$k$ = 4, 5 and 6**

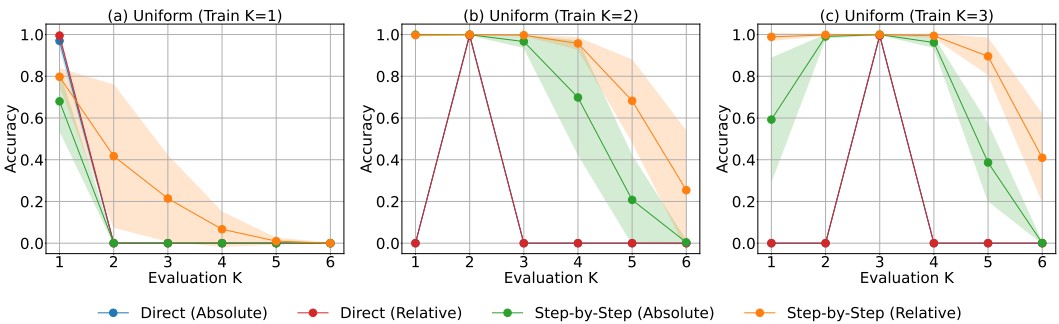

Figure 10: **Cross-k evaluation (Uniform), train-$k$ = 1, 2 and 3**

A.7 COMPOSITION EQUIVALENCE AND SHORTCUT LEARNING IN STEP-BY-STEP MODELS

**Percentage of samples where step-by-step models exhibit shortcut learning**

In Figures 4b(a) and (b), we observe that step-by-step models have higher accuracy if evaluated only on the final output vs. evaluated on full output (including intermediate outputs). As we measure sharp accuracies (i.e., 1 for getting the complete answer correct, zero otherwise), the difference between these accuracies also corresponds to the percentage of samples on which models exhibit shortcut learning. In 13, we show this difference and can observe that for some splits (e.g., size = 200), models exhibit shortcut learning for over 50% of samples.

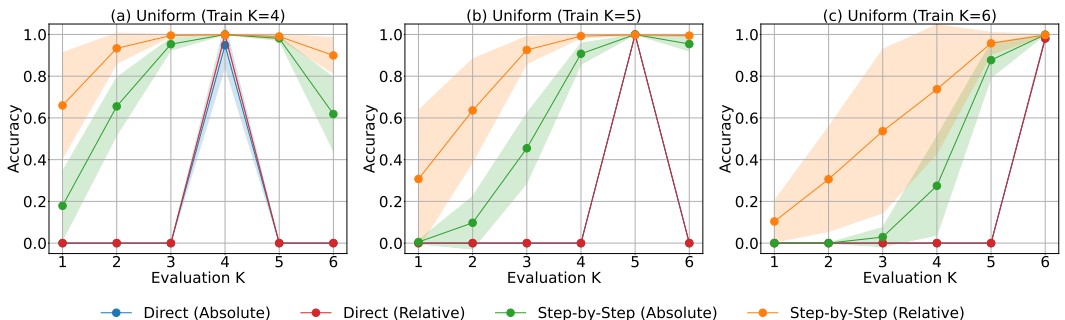

Figure 11: **Cross-k evaluation (Uniform), train-$k$ = 4, 5 and 6**

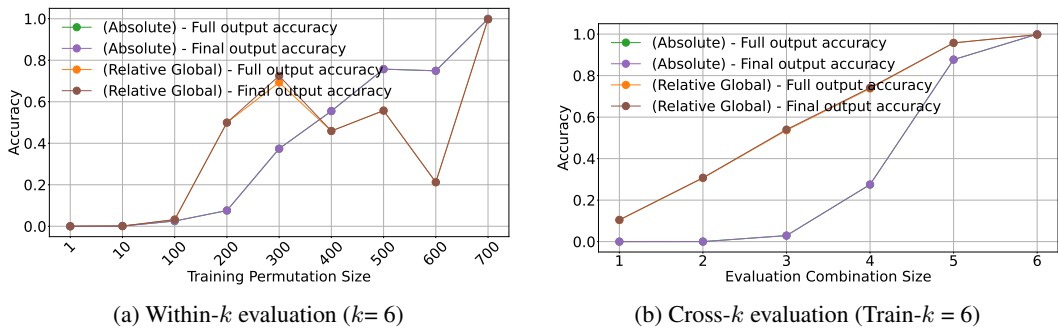

(a) Within-$k$ evaluation ($k$= 6)    (b) Cross-$k$ evaluation (Train-$k$ = 6)

Figure 12: **Eliminating composition equivalences removes shortcut learning in step-by-step models :** We observe that the lack of composition equivalence in the uniform benchmark removes shortcut learning behavior in the step-by-step models. This is shown by the exactly same step-by-step accuracy, and the final accuracy is the same for both within-k evaluation ($k = 6$) and cross-k evaluation (train-$k = 6$).

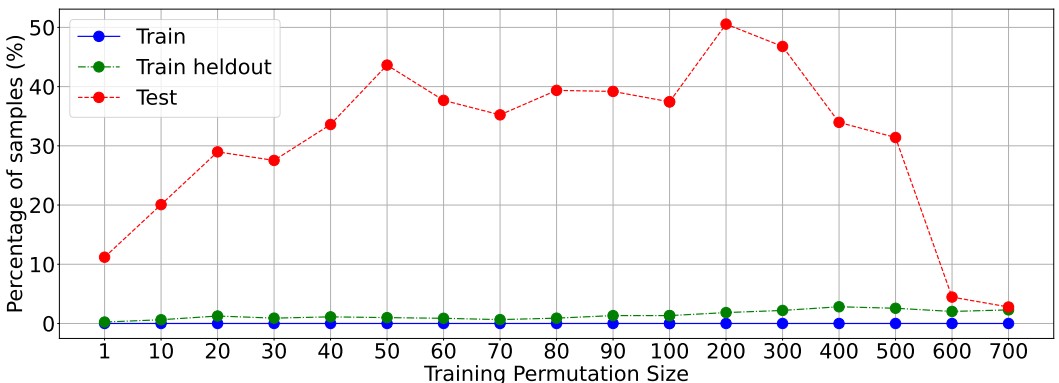

Figure 13: **Percentage of samples where step-by-step models exhibit shortcut learning**

## A.8  ISOLATING THE EFFECT OF COMPOSITION EQUIVALENCE ON THE PERFORMANCE OF DIRECT MODELS FOR THE DIVERSE BENCHMARK

### A.8.1  SHARED VS. NON-SHARED EQUIVALENCE CLASS EXAMPLES

In Figure 15, we show two equivalence classes of size 38 (ID 9) and 24 (ID 2) for $K = 6$. First equivalence class usually ends with (`'union'`, `'map'`) or (`'join'`, `map`) sequences, and the second class ends with (`'union'`, `'join'`), causing end-to-end behavior difference between the two classes. We also observe that the equivalence scores vary significantly within an equivalence class, ranging from exact equivalence (1.0) to weak equivalence (0.01). Next, we observe that the

model performance is significantly higher for test sequences that belong to equivalence classes with shared members in the train set and remains poor (0-0.2) for sequences that belong to non-shared equivalence classes.

### A.8.2 SYSTEMATICALLY INCLUDING TEST SEQUENCES WITHOUT SWAPPING WITH TRAIN SEQUENCES (K=6)

In Figure 14(a), we show that the generalization performance increases with shared equivalences in the case we systematically leak half of the members of a test equivalence class in training without swapping. This reduces the test split size in proportion to the class size as the percentage of shared equivalences increases. Figure 14(b) shows that performance increases significantly for shared equivalence classes while performance of non-shared equivalences remains poor, varying between 0%-20%.

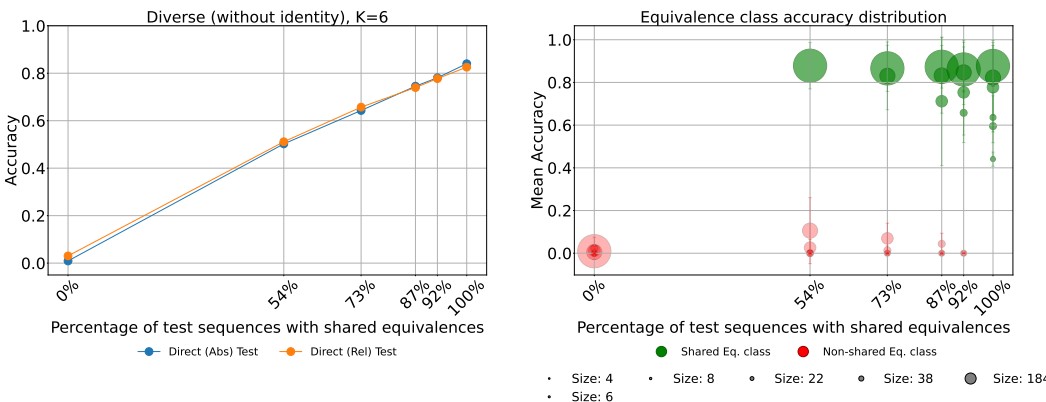

Figure 14: **Leaking test equivalence classes without swapping (K = 6):** (a) Generalization performance to unseen sequences is correlated strongly with the number of shared composition equivalence classes between train and test. (b) Mean accuracy of shared equivalence classes is significantly higher than that of non-shared equivalence classes.

### A.9 GENERALIZATION PERFORMANCE OF LARGER MODELS

### A.9.1 COMPOSITIONAL GENERALIZATION EVALUATION OF PRE-TRAINED MODELS

We fine-tune and evaluate compositional generalization performance of pre-trained Gemma3-1B models on the following train/test splits. Fine-tuning is done for five epochs with a batch size of 1.

- Diverse (without identity, K=6, random 80/20 split): Train acc (heldout): 96%, **Test acc: 1%**.
- Diverse (without identity, disjoint split with 0% shared equivalences): Train acc (heldout set): 97%, **Test acc: 93%**.

We consider these splits as they represent extreme settings to validate the hypothesis that composition equivalence also affects the generalization of larger pre-trained models. We focus mainly on direct models, as composition equivalence primarily affects them, and fine-tuning across all the systematic splits and models considered in this paper is computationally intensive.

Figure 17 and 18 show the accuracy distribution over equivalences, and we can see that all classes are shared in the random 80/20 split. In contrast, in Figures 19 and 20 in the disjoint split, no classes are shared and accuracy distribution of classes is pretty low.

### A.9.2 LARGER TRANSFORMER ARCHITECTURE (N_HEADS = 12, N_LAYERS=12)

In Figure 21, we observe that the performance of the larger architecture shows similar trends seen in Figures 2(b) and (d) corresponding to the smaller architecture (N_heads = 6, N_layers = 3).

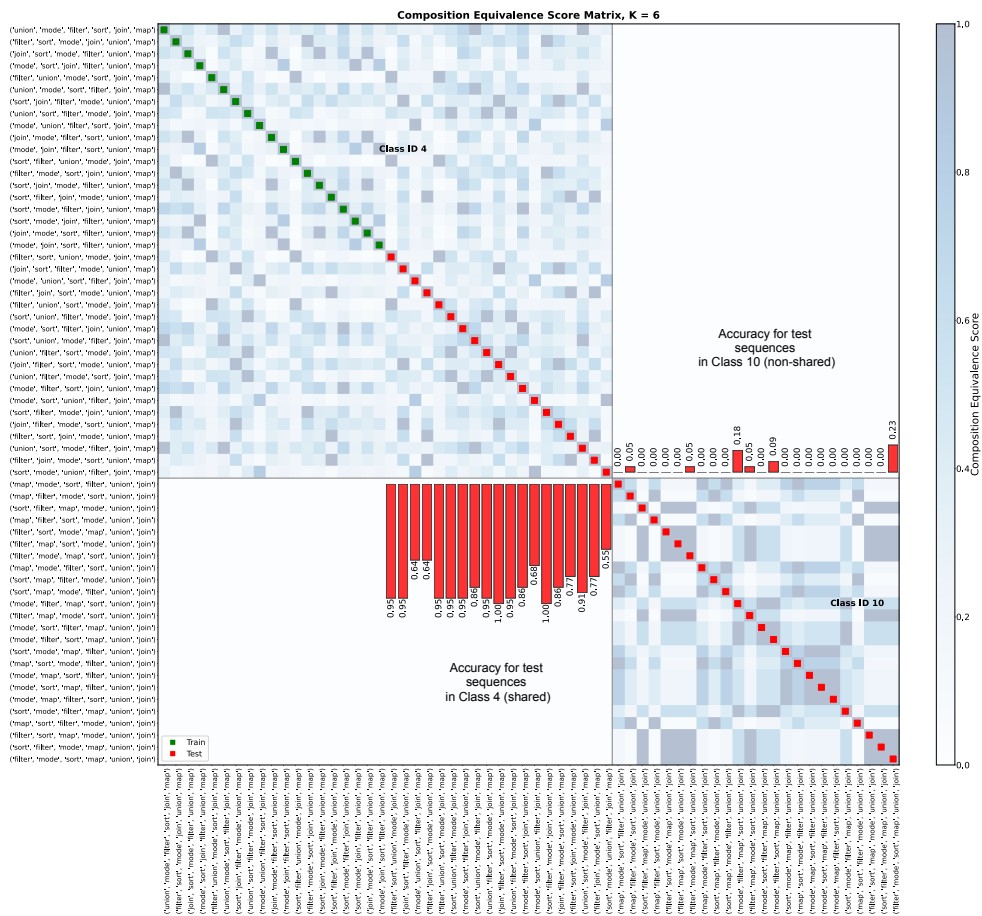

Figure 15: **Visualization of learned equivalence classes for the diverse benchmark (K=6):** We show two equivalence classes (out of a total of 14 classes). First, we can note that the degree of equivalence varies significantly within an equivalence class. The test sequences are marked in red, and the training sequences are marked in green, indicating whether an equivalence class is shared. We observe that accuracy is significantly higher for compositions in the shared equivalence class and near zero for those in the non-shared equivalence class.

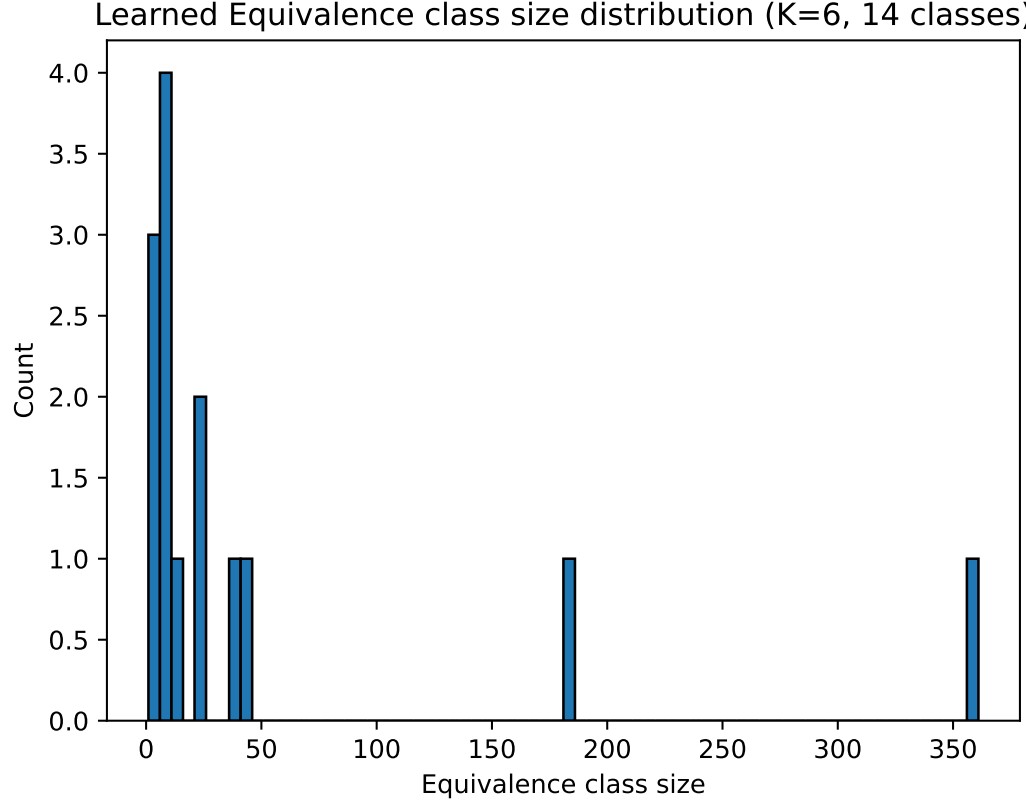

Figure 16: **Size distribution of learned equivalence classes(K = 6)**

A.10   REPRESENTATION VISUALIZATION OF EQUIVALENCE CLASSES

To analyze the internal representations learned by our models, we extract hidden states from the final layer normalization block of the transformer. During inference, we hook the final layer block to capture the hidden representation associated with the last generated token at each decoding step. For each input sequence, this yields a fixed-dimensional vector summarizing the model's processing of the prompt and its continuation. We perform this extraction across both training and held-out test sets. To study the structure of these embeddings, we apply t-SNE to reduce the dimensionality of the representation matrix to two dimensions.

In 22 and 23, we present t-SNE plots of the two-dimensional representations of all training and test samples, where squares and diamonds denote training module orderings and circles denote test orderings. Test sequences are colored according to their evaluation accuracy. Training orderings that are deemed equivalent to a given test ordering are highlighted on a blue scale, with intensity determined by their *equivalence class score*. Formally, let $f^{\text{test}}$ denote a test function composition, $f^{\text{train}}$ a training composition, and $x$ an input string. Denoting the model output as $\hat{y}(f, x)$, the equivalence class score is

$$S(f^{\text{test}}, f^{\text{train}}) \;=\; \sum_{x \in \mathcal{X}} \mathbf{1}\big[\hat{y}(f^{\text{test}}, x) = \hat{y}(f^{\text{train}}, x)\big]\,,$$

where $\mathcal{X}$ is the set of test input strings and $\mathbf{1}[\cdot]$ is the indicator function. Thus, the score reflects the number of inputs on which the model assigns identical outputs to the two function orderings.

From the figures, we demonstrate visually what we present in the main paper, that direct models only demonstrate performance when there exist equivalence classes in the uniform dataset. We can see in 22 that the outputs for high accuracy test orderings are only those which predict the same output as the train equivalence classes. Upon investigating the diverse dataset in 23, we find that the

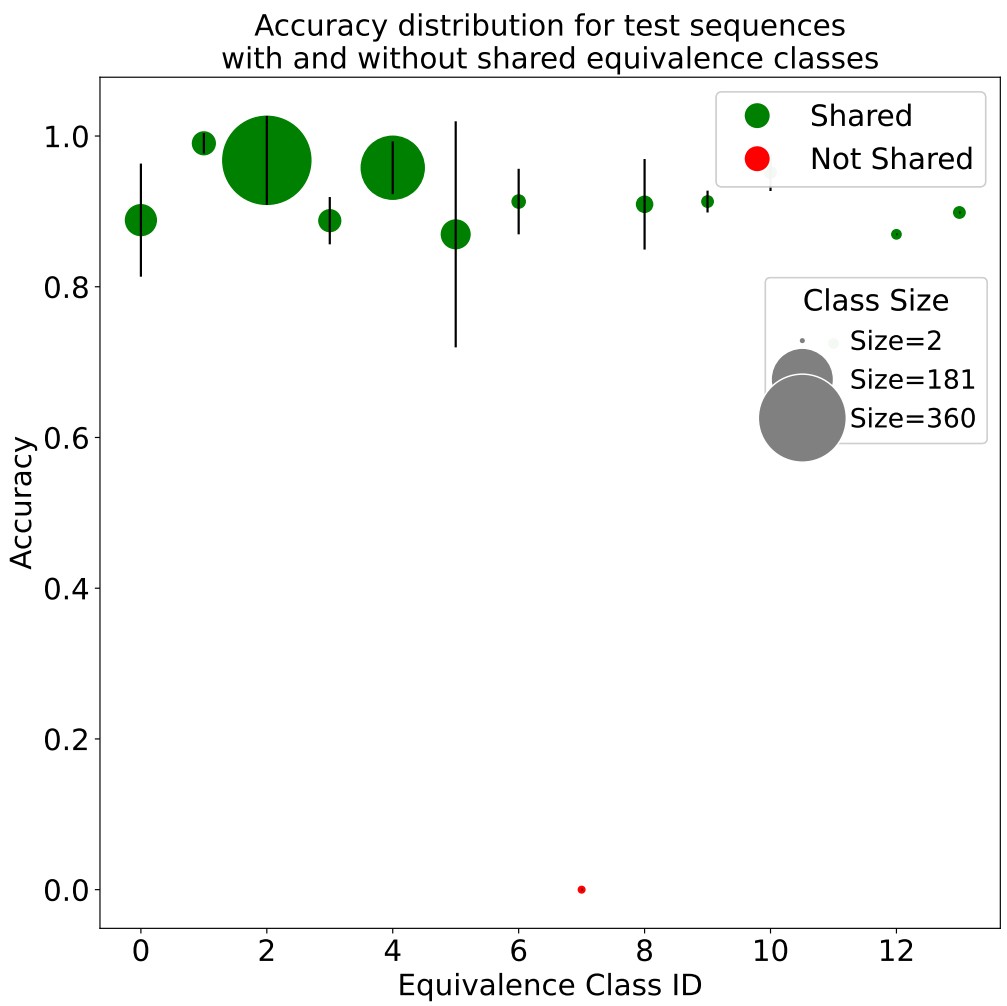

Figure 17: **Gemma-1B performance distribution with random 80/20 split:** We observe that most of the equivalence classes are shared and the model has overall high accuracy. Only one class is not shared where the model has low accuracy.

patterns are less identifiable, likely because of the many equivalences that exist within the training data.

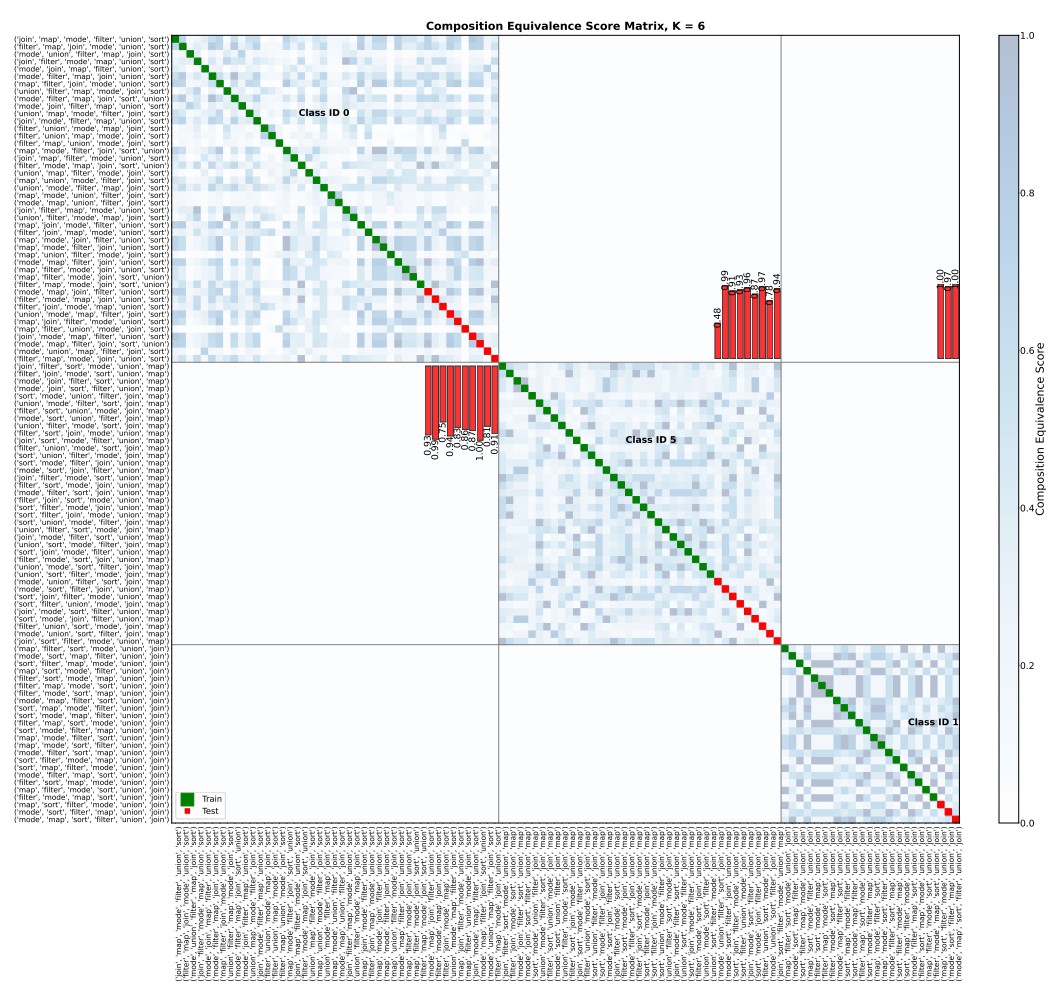

Figure 18: **Gemma-1B visualization of per-class distribution for 80/20 split:** We observe that most of the equivalence classes are shared and the model has overall high accuracy over test sequences in the shared equivalence classes.

## A.11 PREVIOUS MODULE COVERAGE EXPERIMENTS

In this section, we present module coverage results from Section 4, plotted against the test splits for random and systematic sampling of compositions.

**Effective training of step-by-step models requires much smaller training sets with random sampling than with systematic sampling:** Figure 24(a) shows that step-by-step models achieve higher compositional generalization performance after seeing only 10 (1%) sequences with relative position embeddings and roughly 100 (14%) sequences with absolute embeddings. However, with systematic selection (Figure 24(b)), models need to see a larger number of sequences. We also observe a drop in performance at $n = 600$ permutations (80%). Upon further analysis, we find that the poor performance for $n = 600$ is due to the test set consisting only of compositions starting with $f_6$, which none of the systematic orderings in the training set had, demonstrating effects of module coverage failure.

In the case of composition equivalences in the diverse benchmark, step-by-step models need more sequences under random selection than needed in the uniform benchmark without equivalences (Fig-

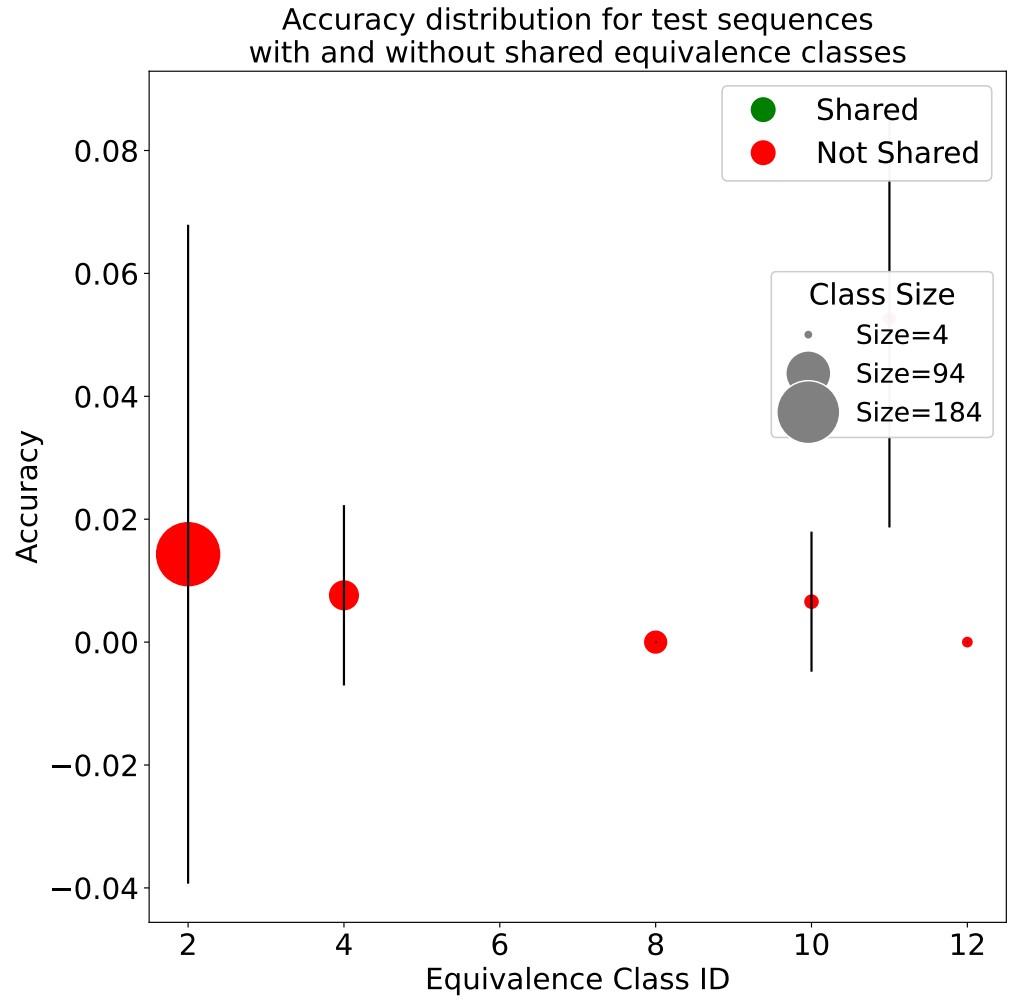

Figure 19: **Gemma-1B performance distribution on disjoint test split (no shared equivalences):** We observe that the model has poor test accuracy (1% mean) as no test equivalence class is shared in the training set.

ure 24(c)). This is due to composition equivalences at the intermediate output level, which creates shortcut learning in step-by-step models, as discussed in Section 3.

**Module coverage interacts with composition equivalence in direct models:** For the diverse benchmark, 24(c) and (d), we observe that these models saturate at a lower performance in the case of systematic composition selection than in the case of random selection, showing that module coverage also affects the learning of composition equivalences. The difference in performance due to the varying module coverage between train-test shows that merely accessing intermediate outputs for step-by-step learning is *insufficient* for models to exhibit robust compositional generalization, and that module coverage affects learning of equivalences in direct models.

### A.12   FAILURE ANALYSIS

#### A.12.1   WITHIN-$k$ EVALUATION

**Step-by-step models for $K$=2, and diverse benchmark** n Figure 2(b), both direct and step-by-step models perform poorly when evaluated on tasks with k=2 modules. This happens for two main

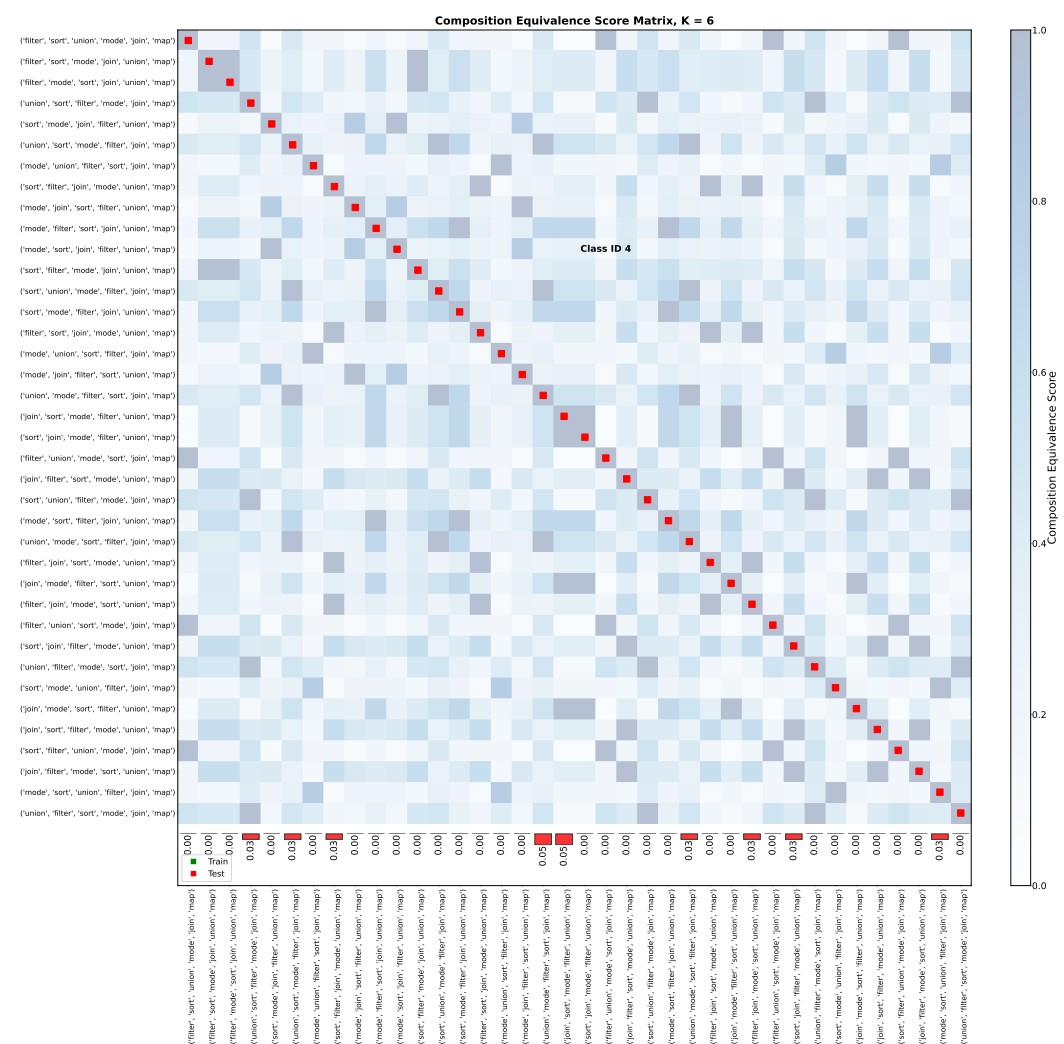

Figure 20: **Gemma-1B visualization of test accuracy for the split with no shared equivalences:** We observe that none of the equivalence classes are shared, and the model has poor accuracy over test sequences in those classes.

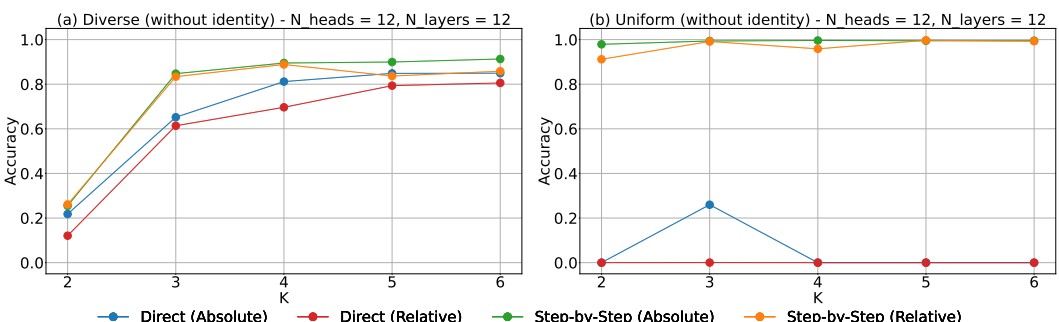

Figure 21: **Generalization performance of bigger transformer architecture**: We observe that the performance of the bigger architecture shows similar performance trends as seen for the smaller architecture (N_heads = 6, N_layers = 3) in Figures 2(b) and (d).

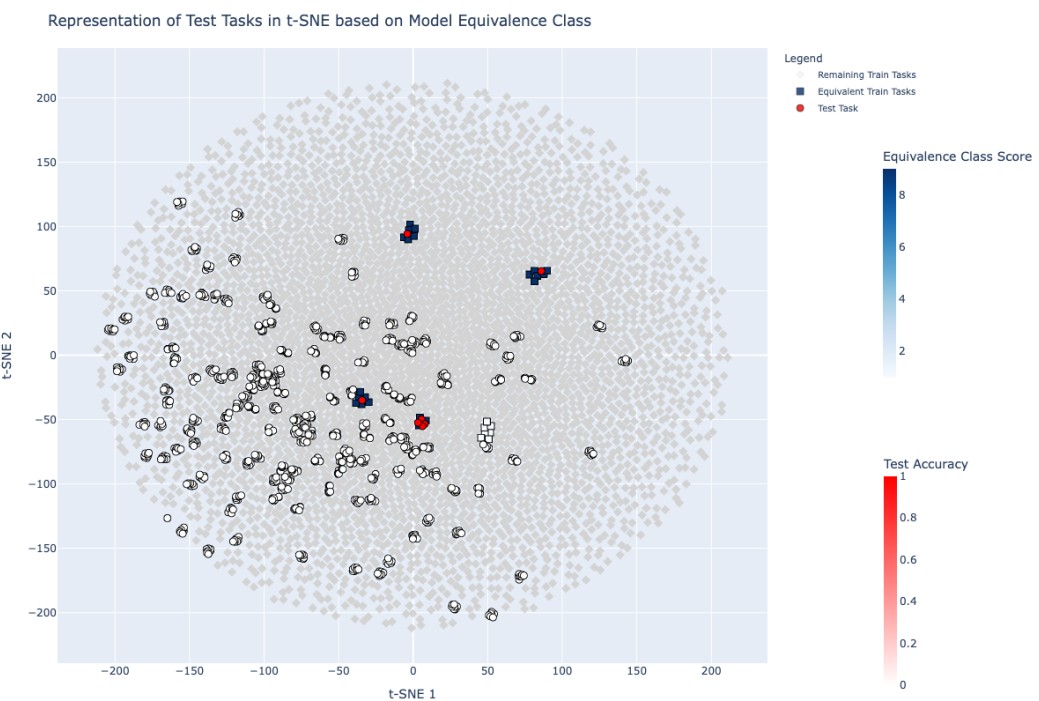

Figure 22: **TSNE representation of model evaluated on uniform benchmark with four equivalence classes shared:** We highlight test sequences and the corresponding equivalent training sequences. Test sequences are colored based on their accuracy (Dark red: 1.0 accuracy and white: 0.0 accuracy). In this split, four identity-based equivalence classes are shared between train and test, and we can see that test sequences only belonging to those equivalent classes have 1.0 accuracy, while remaining test sequences have 0.0 accuracy, demonstrating the equivalent class phenomenon at the model representation level. For the purpose of this plot, we evaluated the model on the same input data tokens to visualize equivalences at the final layer output representation level. In our actual experiments, we sampled distinct input data tokens.

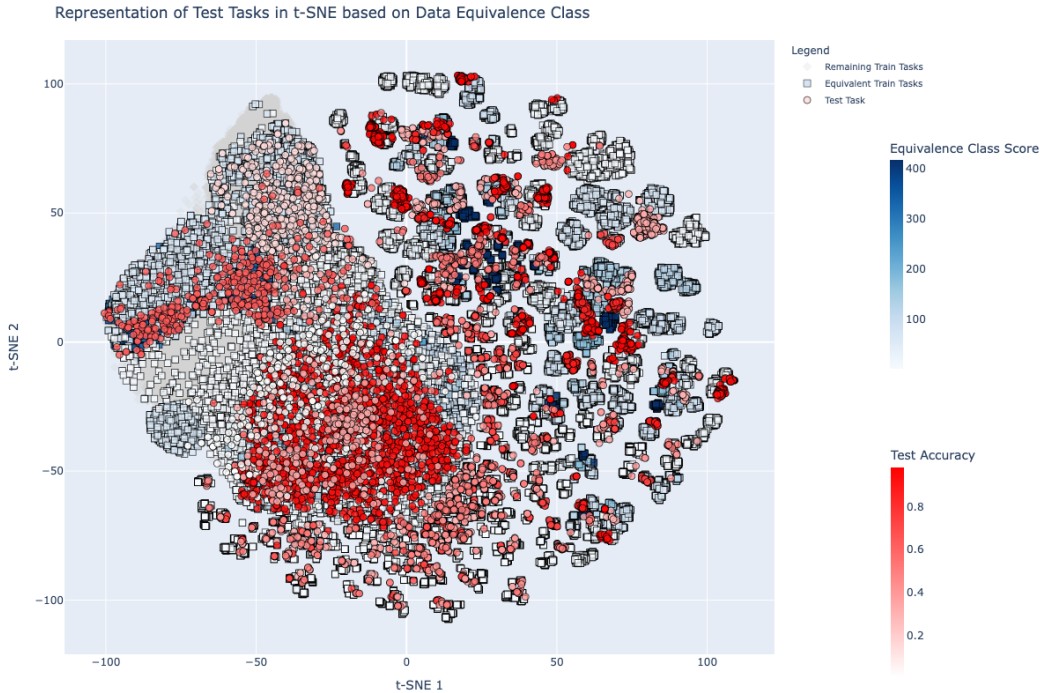

Figure 23: **TSNE representation of model evaluated on diverse benchmark with $k = 3$:** We highlight test sequences and the data-generating process based on the computation of composition equivalence. Test sequences are colored based on their accuracy (Dark red: 1.0 accuracy and white: 0.0 accuracy). Overall, we can see that diverse benchmark has a wide variety of equivalences among different sequences. There also exist approximate equivalences based on the input strings. For example, small clusters on the right correspond to the single-character outputs resulting from the `mode` and `filter` operations. Multiple sequences can belong to different equivalence classes, depending on the input data.

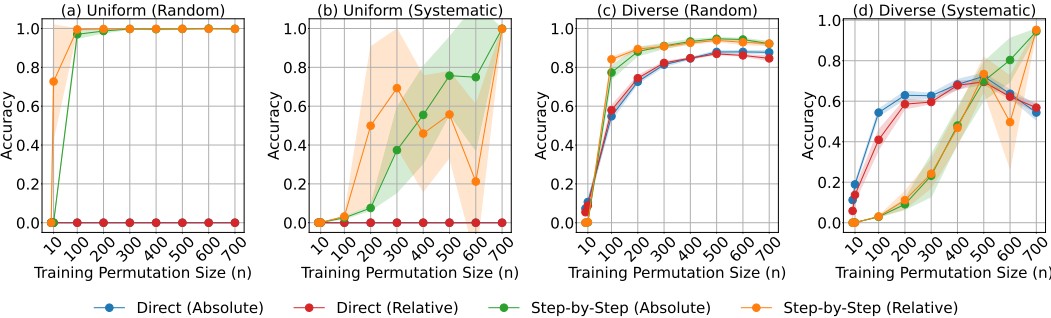

Figure 24: **Random vs. systematic selection of compositions:** (a,b) Step-by-step models generalize faster with random selection due to reduced spurious correlations between positions and task tokens. Systematic selection creates position-task correlations that slow generalization. (c,d) Random sampling converges faster than systematic sampling. Direct models achieve lower performance with systematic selection than with random selection.

reasons: First, there are fewer possible task combinations without identity modules. Without identity modules, there are only 30 unique task sequences, but with identity modules, there are 630 sequences (when $k_{max} = 7$ and $k = 2$). This means models with identity modules get a larger number of training sequences. Second, identity modules create beneficial mathematical relationships between different task combinations, which helps improve overall performance. Similar logic applies for k=3. With k=4, 5, the number of training sequences increases to 360 and 720, respectively.

### A.12.2   EQUIVALENCE CLASS NECESSITY EXPERIMENT

We focus on sequences with $k = 6$ non-identity functions and one identity function, generating a total $7! = 5,040$ permutations grouped into a total of 620 equivalence classes. Each class consists of seven equivalent tasks corresponding to the possible positions of the identity module, while keeping the relative ordering of non-identity functions fixed. We maintain constant train-test split sizes in terms of equivalence classes (576/144) and vary the percentage of shared equivalence classes from 0–100%. To maintain a fixed total number of samples, we exchange half of the tasks within shared equivalence classes between the training and test sets. A setting of 0% means disjoint classes, while 100% means all test tasks have equivalents in training. Accuracy is computed based on shared equivalence classes.

**Failure details:**

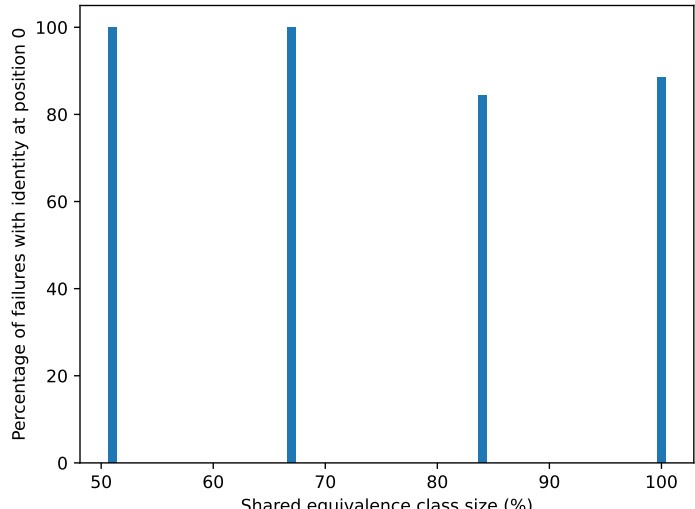

Figure 25: **Percentage of test sequences with 0.0 accuracy and identity at first position**

