# OpenReview forum: "Why Transformers Succeed and Fail at Compositional Generalization: Composition Equivalence and Module Coverage"
_ICLR.cc/2026/Conference — Submitted to ICLR 2026_

### Official Review · Reviewer_tN2B · 2025-10-31

**Soundness:** 3
**Presentation:** 4
**Contribution:** 3
**Rating:** 6
**Confidence:** 4

**Summary:**

The authors evaluate Transformers trained from scratch on a series of sequence learning tasks that require compositional generalization. They propose two theoretical criteria (composition equivalence and module coverage) that they suggest explain generalization in different variants (step-by-step vs. direct output and uniform vs. diverse sampling). In particular, this provides a systematic explanation for generalization patterns that at first seem idiosyncratic and provides important context for prior findings of successful compositional generalization.

**Strengths:**

I thought this was an interesting paper. It is well-written and easy to follow. The theoretical setup is clear, well-motivated, and interesting. Similarly, I think the theoretical definition of composition equivalence is really valuable and I appreciated the effort to identify fundamental theoretical principles that predict compositional generalization. Overall, I think that this is a thought-provoking paper that dives deeply into disentangling different factors contributing to compositional generalization (or the lack thereof) and will be valuable to the field. Below are a few things I liked in particular:
- I thought the introduction compactly provided a great motivation for the investigation.
- Figure 1 was very helpful.
- I think your identification of the potential for shortcut learning is very helpful to keep in mind; it provides evidence in a simple setting for something that machine learning researchers have articulated as a worry in a naturalistic setting as well: just because models output their reasoning step-by-step doesn't actually mean that they *use* that reasoning.
- Figure 5 is very insightful and provides important validation for your theory (and indeed it would be very valuable to go a little further in this evaluation, see below).

**Weaknesses:**

As noted, I think this is a very interesting paper. However, I think it would benefit from a closer investigation/validation of the theoretical principles it proposes:
- In particular, I think some of the explanations on how the proposed principles explain the observed generalization patterns could have been more specific (see section 1 under Questions).
- I think your proposed principles should allow you to predict the specific subsets of the test data for which models should generalize or not generalize (in my mind, this would be particularly interesting on the diverse dataset without the identity mapping). I think this would provide stronger evidence for your proposed mechanism and would also nicely extend your investigation in Figure 5. In particular, this would provide insight into whether the same relationship between the number of equivalences and accuracy holds for the diverse dataset, which would be an important step towards scaling these principles towards realistic data. (See section 2 under Questions.)
- Module coverage currently lacks a formal theoretical definition. While I think your investigation in Section 4 is very interesting, I think having such a formal definition would be very helpful in providing a basis for future extensions of your investigations (much like your formal definition of composition equivalence is very helpful in this direction). (See section 3 under Questions.)

Overall, I want to emphasize that I really appreciate the approach in this manuscript: I see its primary contribution as a systematic investigation of principles of compositional generalization, which I think is very valuable. In light of this primary contribution, I think it would be useful to further solidify this systematic investigation. While I think the manuscript in its current form would be a worthy addition to ICLR, addressing the weaknesses above would substantially improve the paper and cause me to further increase my score.

**Questions:**

**1 Clarifications about composition equivalence**
- In l. 330-340, I think it would be valuable to make clear that the six bijections you sampled don't allow for any other composition equivalence (since there would be bijections for which this wouldn't be true, e.g. if two of the bijections were each others' inverses). I think that is what is implicit in your statement that there are no other composition equivalences, but I don't think that's a trivial statement. I imagine that it's just super unlikely under random sampling and this is very intuitive for, say, concatenations of two functions. However, it is less intuitive to me how the probability of two different concatenations of functions being equivalent scales with the number of times $k$ that you're concatenating them. I imagine for $k=2$ it's close to zero, for $k\to\infty$ it converges to $1$. Is it still close to zero for e.g. $k=6$? My sense is probably yes, since there are 26! possible bijections but only $6^6$ possible concatenations of the six bijections you sampled and 6^6 is vanishingly smaller than 26!. But it would be helpful to expand a bit on this point, e.g. through a formal argument or by brute-force checking the specific bijections you sampled. (Apologies for the long bullet point for a relatively minor point.)
- Similarly, the statement in l.325-327 is not obvious to me. Does random sampling necessarily create train-test splits consisting of composition equivalences/what is the likelihood of creating it?
- Why does identity-based composition equivalence exist only among sequence lengths with fixed $k$? Isn't $f_1\circ f_2\circ \text{Id}$ equivalent to $f_1\circ f_2$? I understand that that does not qualify as composition equivalence according to your definition (which requires the same $k$ for $F$ and $F'$), but why does the two functions would still be identical, so why is that unable to support compositional generalization?

**2 Evidence for the role of composition equivalence**
- Could you split the test dataset in the diverse setting without identity into test data where the training data contains equivalent functions and where it doesn't? Your theory would predict that for data where the training data contains no equivalent functions, accuracy should be zero, correct? Similarly, would this allow you to create a plot where you plot accuracy as a function of data points in the training set with an equivalent function?

**3 Module coverage**
- Could you speak to how you would think about a formal definition for module coverage?
- Even better, do you have any empirically testable predictions for which specific permutations the models should and should not generalize to? (Expanding on your observation in l. 458-461.)

**Other questions/suggestions**
- I think having an example input-output pair somewhere in the main text would have been helpful during task explanation (though your explanation was still pretty easy to follow). It would also allow you to visually depict
- I don't understand why your mapping always outputs a sequence of the same length $m$. filter, e.g. should decrease the sequence length, no?
- You may find it interesting to relate shortcut learning to the literature on faithfulness (e.g. https://arxiv.org/abs/2307.13702), which attempts to measure whether language models actually rely on their step-by-step reasoning. My understanding is that shortcut learning provides an instance where reasoning is not faithful.
- While I don't think this paper requires evaluation on real-world benchmarks (and indeed benefits from the toy setting), do you have any thoughts on how composition equivalence and module coverage relate to real-world settings and perhaps some observed empirical phenomena?
- Do you have any thoughts on the minimal setting where researchers could investigate your results? I.e. how small could we make the set of possible tokens, possible tasks, etc. while still capturing the most important aspects of your insights? I think this would be interesting to potentially ground future theoretical investigations.

**Final note**

I realize that this is a pretty long set of questions, so I want to emphasize that I do not expect the authors to necessarily address all of my questions (especially as some of my suggestions would involve new empirical experiments, which may not be feasible during rebuttal). My questions are intended to provide some specifics on how I think the authors could further concretize their theoretical principles and the evidence for the role they play in compositional generalization. Indeed, I think the many questions this submissions raises shows that it is already quite a thought-provoking paper.

---

> ### Author Response · Authors · 2025-11-23
> **Response to reviewer tN2B**
>
> We highly appreciate the positive, detailed, and thoughtful feedback. We are pleased to know that you found our paper thought-provoking.
>
> **Regarding concern about further evidence on the role of composition equivalence:** Thank you for the great suggestion, and we hope that the newly added results (Figure 6) in the revised manuscript on pages 8,9, and the appendix (also discussed in the combined response above), answer your question about making predictions on specific splits of the diverse benchmark based on the degree of composition equivalence.
>
> **Regarding the formalization of module coverage and empirically testable predictions:** We hope that including a quantitative metric in Section 4 addresses your concerns. This metric quantifies the extent to which a test set is out-of-distribution in terms of coverage. Results in Figure 7 show that performance can be reliably predicted at the extremes—near-perfect at low divergence values and near-zero at high divergence values—but remains variable at intermediate levels for both direct and step-by-step models. This is characteristic of any out-of-distribution evaluation in general: there inherently exists a range of distribution shifts where model performance becomes unreliable and hard to predict without strong assumptions.
>
> **Implications for real-world settings and observed phenomenon:** The sequential composition tasks involving various string manipulation functions considered in the paper are inspired by code-output reasoning in standard coding benchmarks, such as [CRUXEval](https://crux-eval.github.io/). It is observed that bigger models, such as GPT-4, can show up to 63% accuracy on code output reasoning. Decomposing the performance of these models due to the presence of equivalent programs during training (e.g., different implementations of the same program) is important for understanding their true reasoning capabilities. Another real-world implication is that when models exhibit shortcut learning, our work shows that composition equivalence is one of the factors. Making changes at the data level to reduce these equivalences, e.g., by including fine-grained data, is one possible remedy. However, there might also exist scenarios where having equivalences in the data is helpful for making the model learn certain capabilities—for example, if our goal is to reduce context length and we want the model to do well on unseen complex compositional tasks, including tasks with similar behavior would help.
>
> **Clarifications about composition equivalence:** Thanks for the excellent observation about the low likelihood of the existence of composition equivalence in the case of random bijections with a large vocabulary size. We mainly computed the equivalence scores among the sequences by evaluating them over a large number of strings (1000) and found no match among them. And when we included identity and performed identity-based equivalence experiments, we found only identity-based equivalences. We appreciate the calculation you have provided.
>
> **Random sampling and composition equivalence:** Random sampling in the case of including the identity module creates a large number of id-based equivalence class with a large number of members in it. For example, in the case of max sequence length 7, total non-identity modules = 6, id-based equivalence class size is 21 (K=2), 35 (K=3), 35 (K=4), 21 (K=5), and 7(K=6). This counts various permutations of the identity module at different positions (treating them as identical). With 80/20 sampling, there is a probability of roughly 17, 28, 28, 28, and 6 samples going into training and the remaining into test, providing a sufficient number of examples to learn equivalences.
>
> **Identity-based composition equivalence exists only among sequence lengths with fixed $k$:** In the paper, k$ denotes the number of *non-identity* modules. Thus, your example also has a fixed k. If the number of non-identity modules changes in the train/test split for random bijections, end-to-end behavior changes, and hence composition equivalences are less likely to exist. We will make this clearer.
>
> **Answers to other questions**
> - We pad the output with space tokens to keep the output length the same to account for the mode and filter functions as mentioned in Appendix A.3.2. We will clarify this in the main text.
>
> We highly appreciate the suggestions and pointers and would keep them in mind for future revisions.

---

> > ### Comment · Reviewer_tN2B · 2025-11-24
> >
> > Thank you for your response! I think the metrics you're providing and the impact they have an generalization performance improve the rigor of your claims. The non-monotonic curves in Figure 7 are a bit surprising --- while I agree that the more important evidence of your claims is given by looking at the extremes, do you have any thoughts on origins of that non-monotonicity/do you think it's an artifact of your sampling procedure or something real?

---

> ### Author Response · Authors · 2025-11-25
> **Response to the reviewer tN2B**
>
> Thank you for this thoughtful question. The non-monotonicity in Figures 7(b), (d) is primarily at the point 2.12, corresponding to the training permutation size (n)=600 case, where module f6 **never** appears at the first position in the training set but **always** begins the test sequences, causing a high divergence value of 2.12. At the same time, n=700 just has the value of 0.85 (the leftmost point).
>
> With increasing training permutations in the systematic example, we observed a decrease in the divergence metric, except for n=600, which is an exception due to this particularity.
> Thus, this particular discontinuity is an artifact of our sampling procedure because it happens that two adjacent datapoints, 2.12 (n=600) and 1.73 (not marked for clarity, n=200), showcase different failure modes with close divergence values, but the fact that models are failing for both points (either with low mean or high variability) is an indication of the underlying phenomenon of lack of coverage.

---

### Official Review · Reviewer_ya87 · 2025-11-01

**Soundness:** 2
**Presentation:** 2
**Contribution:** 2
**Rating:** 4
**Confidence:** 2

**Summary:**

This paper investigates why transformer models succeed or fail at task-based compositional generalization by looking into the ability to combine known modules (e.g., sort, filter) into novel sequences. The authors identify two key data-centric factors: composition equivalence and module coverage, showing that compositional generalization in transformers is heavily influenced by the data-generating process, not just model architecture, and calls for more careful benchmark design to avoid evaluating shortcut learning instead of true reasoning.

**Strengths:**

1. The paper introduces and formalizes composition equivalence and module coverage as key factors influencing compositional generalization, shifting the focus from model-centric to data-centric explanations.

2. The paper uses controlled synthetic benchmarks (uniform vs. diverse functions) and systematic train-test splits (within-k vs. cross-k) to cleanly isolate the impact of each factor.

**Weaknesses:**

1. The failure results from using step supervision seem confusing and lacks more in-depth discussion. There could be multiple factors: Does the model still choose to ignore the intermediate result even in training when such supervision is given, or does it happen at test time? How often is composition equivalence still maintained even in step level supervision? There is no experiments isolating the factors.

2. Experiments are conducted only with small GPT-2-style models. It remains unclear whether the findings generalize to larger, more capable transformers or real-world tasks.

3. The shortcut learning problem introduced in this paper does not seem to be specific to Transformers. I can imagine it happens for other model architectures as well since it is more of a data problem, so the contribution is not very clear.

**Questions:**

Do you expect these findings to hold at scale? Would larger models learn more robust compositional reasoning, or just more sophisticated shortcuts?

---

> ### Author Response · Authors · 2025-11-22
> **Response to reviewer ya87**
>
> We thank the reviewer for the valuable feedback.
>
> **Regarding failure results from using step supervision:**
> Thank you for the great questions. We have included additional analysis in Appendix A.7 (e.g., Figure 13). Please find the answers below.
> Shortcut learning behavior is mainly observed in model evaluation on unseen test compositions. In Figure 13, we show the % of samples for which the model exhibits a difference between final and full output accuracies. Specifically, we observe no difference in the training data set. For the train heldout data set (i.e., data set with the same compositions as in the training data but consisting of different input strings), we observe a slight difference of up to 2%. For the test set, this difference is significant across certain splits, up to 50%, as also shown in Figure 4(a). Thus, the more out-of-distribution evaluation gets along with the presence of functions allowing shortcut reasoning (e.g., mode, filter), the more we observe the shortcut-learning behavior of step-by-step models.
>
> **Regarding generalization to larger models**: As we mentioned in the common response, we performed additional experiments on Gemma 1-B models and larger transformer architectures, and we show that some of the findings related to shortcut learning, e.g., generalizing by learning composition equivalence applies to pre-trained models as well as larger transformer architectures as well as  (Figures 17-21, Section A.9).
>
> **Regarding the specificity of the results to the Transformer architecture:**
> We focus on transformer architecture because of its widespread use in LLMs and many other applications, such as vision and biological models. A few prior studies (e.g., Hupkes et al., 2020; Ramesh et al., 2024) also evaluated the compositional generalization abilities of other sequential ML models, such as RNNs/LSTMs, but found that transformer-based models outperform them. So, we wanted to focus on models where we already observe success, and our goal is to understand the mechanisms behind transformer success and the role different data distribution shifts play in it. Studying shortcut learning in other model architectures is an interesting question, but beyond the scope of this work.
>
> **Motivation for using synthetic benchmarks:** We focused on synthetic tasks to allow systematic control over train/test splits while keeping model size smaller. As we move to more complex real-world tasks, the model size requirements also increase, making it computationally challenging to run a large number of systematic experiments, as done in this paper, to understand the role of key factors comprehensively. Another reason to focus on simple string manipulation tasks is that we wanted to focus on functions that can be represented and learned by the Transformer architecture easily (e.g., RASP primitives; Weiss et al., 2021). This design choice is inspired by a large body of work studying out-of-distribution characteristics in Transformer architecture using various mathematical and algorithmic reasoning tasks, e.g., length generalization studies (Anil et al., 2022; Zhou et al., 2024).
>
> **Implications on real-world tasks:** The sequential composition tasks considered in the paper are similar to code-output reasoning benchmarks, such as [CRUXEval](https://crux-eval.github.io/). It is observed that models such as GPT-4 can achieve up to 63% accuracy in code output reasoning. Decomposing the performance of these models due to the presence of equivalent programs during training (e.g., different implementations of the same program) is an important future direction of this work.
>
> References:
> - Hupkes, Dieuwke, et al. "Compositionality decomposed: How do neural networks generalise?." Journal of Artificial Intelligence Research 67 (2020): 757-795.
> - Anil, Cem, et al. "Exploring length generalization in large language models." Advances in Neural Information Processing Systems 35 (2022): 38546-38556.
> - Zhou, Hattie, et al. "What Algorithms can Transformers Learn? A Study in Length Generalization." The Twelfth International Conference on Learning Representations.

---

### Official Review · Reviewer_8EDx · 2025-11-01

**Soundness:** 3
**Presentation:** 3
**Contribution:** 2
**Rating:** 4
**Confidence:** 4

**Summary:**

This paper studies the ability of transformers to generalise compositionally by studying the effects of two properties of a dataset on the model's learning and inductive biases. The two properties are composition equivalence -- a property where multiple sequences of computations can lead to the same outcome, meaning that the correct sequence becomes unidentifiable, and module coverage which describes the fact that all modules (or computations) are needed in different orders and positions in the sequence in the training data. The paper shows that if composition equivalence is present in the data or if module coverage is absent then the network will learn shortcuts where computations by the model do not correspond to meaningful (ground truth) manipulations of the data. The work proposes step-by-step modules which also output intermediate steps towards solving a problem as a potential way to mitigate such shortcut learning.

**Strengths:**

## Originality
The work studies and interesting setting for LLMs and considers who important properties of a dataset. There is a wealth of work at the moment aimed at just understanding how transformers perform tasks, however the paper does a good job of justifying why the tasks considered in this work are representative and useful.

## Quality
The hypothesis of the work is clearly stated and the experimental design does seem to isolate the effects of the stated properties of a dataset that should impact prior learning. Clearly, Direct (Relative) does fail in Figure 2 without the identity tokens for example. Results are interpreted fairly overall.

## Clarity
The paper is well written and figures are clear. Captions are also detailed which aids in the clarity of the work. The paper is structured well given the among of content which is covered and so it remains understandable and each section does support the next.

## Significance
Understanding transformers, and in particular their points of failure, is an important line of work given their widespread adoption. The topic of compositional generalisation in transformers in particular is important as this is one of the primary inspirations for the models as language is compositional. Moreover much has been said recently on the emergent reasoning abilities of the models. If shortcut learning is indeed a problem for these models, then this has implicates for a number of related topics like neural scaling laws and these emergent reasoning abilities from scale.

**Weaknesses:**

## Clarity
The paper is quite dense and I find that it remains quite high-level on some of the topics. Some conclusions are also drawn or treated as obvious where I think there are not. The first paragraph of page 7 is particularly problematic for me as much of what is said there is not shown directly but rather implied from the model performance. It is also not explicitly said how the training and validation sets are constructed for this conclusion about removing identity modules and so I am not convinced by the comparison either. Similarly the statement "Cross-$k$ failures occur ...  with fixed $k$" on lines 334 and 335 should also be supported by more direct evidence.

## Quality
My main concern on quality is that the need for additional (and difficult to obtain) supervision is not discussed at all when discussing the step-by-step model. While I appreciate that this work is isolating a phenomenon and is mainly concerned with model behaviour, proposing additional supervision as if it is a good solution (without some consideration for the applicability of the approach) is a bit premature. I will also highlight that my above point on the need to support the conclusions more directly from the results is a quality concern, however I list this under clarity as I think some writing edits could improve upon this.

## Significance
Since the experimental design is one of the primary strengths of the work, the clarity concerns limit my ability to assess significance fully. This is mainly in relation to Ramesh et al. Lines 126 and 127 seem to imply that Ramesh et al. has already considered task order and coverage in this setup and that composition equivalence is the primary contribution of this work. So why not focus more on that? Similarly on lines 339 and 340, why not focus on the clear distinction from Ramesh et al. and the difference with interleaved modules? Overall I would have appreciated a more detailed and clear demonstration of the findings, even at the expense of breadth.

Finally I will note that module coverage has been considered theoretically and it may be helpful to consider this prior work:
- Schug, Simon, et al. "Discovering modular solutions that generalize compositionally." The Twelfth International Conference on Learning Representations.

**Questions:**

How does the paper presented here explicitly build of Ramesh et al.

How would future work aim to practically incorporate the step-by-step modules given the need for additional, detailed supervision.

I would be open to increasing my score and advocating acceptance if these questions are answered and indeed I have missed something fundamental.

---

> ### Author Response · Authors · 2025-11-23
> **Response to reviewer 8EDx**
>
> We thank the reviewer for their efforts and helpful feedback.
>
> **Relevance of work related to Ramesh et al. (2023)**:
>
> - We consider the same task and setup used by Ramesh et al. (2023) for the evaluation of compositional generalization. Their work mainly showed that step-by-step models generalize to an exponential number of compositional generalizations, while direct models do not. In this work, we empirically show that failures in direct models are primarily due to the lack of shared composition equivalences between the training and test data sets. The concept of composition equivalence in task-based composition generalization is novel (to the best of our knowledge), and this is the first time its role in causing superficial generalization in models is clearly shown.
>
> - Further, we characterize composition equivalence and module coverage as two important measures of data distribution shifts—that explain substantial variability in compositional generalization performance for both direct and step-by-step models. Through extensive experiments across a wide range of train/test splits that capture varying degrees of these two kinds of shifts, we provide **strong** empirical evidence that model performance varies significantly with these shifts.
>
> - While Ramesh et al. (2023) also discuss module coverage, they mainly focus on the absolute positions, while we consider both absolute and relative position coverage. With new additional results in Section 4 (also discussed in the combined response above), we provide a continuous metric that measures module coverage divergence between test/train and demonstrates its effect on compositional generalization of both direct and step-by-step models (Figure 7).
>
> We thank you for the pointer related to the module coverage, and we will consider it in future revisions.
>
> **Need for step-by-step supervision**: Prior work has shown that step-by-step supervision improves the compositional generalization performance of transformer-based models (Dziri et al., 2023; Ramesh et al., 2023; Abedsoltan et al., 2025).  Step-by-step supervision, also known as chain-of-thought training or scratchpad training, has been found to improve performance across other algorithmic reasoning tasks as well (Nye et al., 2021; Wei et al., 2022). In this work, we focus on understanding the factors that drive compositional generalization in this commonly used training mode.
>
> **More details about training and validation data sets in the case of explaining identity-based equivalence (Page 7 paragraph):**
> We appreciate the feedback and will clarify the findings below. We will improve the text in the future revision.
>
> The training data set consists of a *subset* of possible compositions of size K; the validation data consists of training compositions with novel data inputs; and the test data consists of the remaining compositions and unseen data strings.
> - In the within-k evaluation, we consider sequences with a fixed number of non-identity modules. For example, for K=3, (mode, map, join) and (mode, sort, filter) are two examples.
> - With identity as a dummy token, we pad the sequences with the id token, assuming a maximum sequence length of 7. For example, (mode, map, join) would give these two possible sequences: (mode, id, map, join, id, id, id) and (id, id, id, mode, map, id, join).
> - When we randomly split data into train and test with an identity module, it is possible that (mode, id, map, join, id, id, id) would go in train and (id, id, id, mode, map, id, join) would go in test, while these two sequences are *exactly* compositionally equivalent. This leads to almost-perfect performance for within-k evaluation with identity modules for both uniform and diverse benchmarks.
>
> - But if we remove the identity module, (mode, map, join) would either go in train or test, and there might or might not exist a compositionally equivalent task in test. For example,  (mode, map, union) is one such approximately equivalent task on a subset of inputs. Due to a lack of exact equivalence, we observe a drop in performance on the diverse benchmark without identity modules. In the case of a uniform benchmark, performance drops to exactly zero, as no equivalent function exists without identity modules due to the random logic of functions.
>
> References:
> - Dziri, Nouha, et al. "Faith and fate: Limits of transformers on compositionality." Advances in Neural Information Processing Systems 36 (2023): 70293-70332.
> - Abedsoltan, Amirhesam, et al. "Task Generalization with Autoregressive Compositional Structure: Can Learning from $ D $ Tasks Generalize to $ D^ T $ Tasks?." Forty-second International Conference on Machine Learning.
> - Nye, Maxwell, et al. "Show Your Work: Scratchpads for Intermediate Computation with Language Models." Deep Learning for Code Workshop.
> - Wei, Jason, et al. "Chain-of-thought prompting elicits reasoning in large language models." Advances in neural information processing systems 35 (2022): 24824-24837.

---

### Official Review · Reviewer_Mq38 · 2025-11-01

**Soundness:** 2
**Presentation:** 4
**Contribution:** 2
**Rating:** 4
**Confidence:** 3

**Summary:**

The paper describes an empirical study of when transformers succeed/fail at task-based compositional generalization, introducing two dataset-side factors: composition equivalence (distinct module sequences that induce the same end-to-end mapping) and module coverage (how uniformly modules appear across positions/contexts). The presence of these factors is controlled by considering two types of datasets: one where sequences are generated by uniform random token sampling ("uniform"), and one where the used composition rules are diverse string ops, inducing some composition equivalence ("diverse"). The uniform data acts as the control dataset, as:
(i) it induces bijections; namely, the only way to have composition equivalence in it is by adding "identity tasks" (essentially, task tokens that don't do anything);
(ii) and module coverage is easily controllable by enumerating token permutations and changing the train-test split.
The authors compare direct vs. step-by-step training under within-k and cross-k regimes, reporting large performance swings that they attribute to these two factors.

**Strengths:**

- The topic is timely and relevant, as the paper proposes a deeper analysis of LLM behavior grounded on compositional generalization, which can be seen as a form of human alignment.
- The notions of composition equivalence and module coverage are well formulated and relevant to the study of compositional generalization.
- The experiments are well designed and substantiate the conclusions drawn by the authors to some extent.

**Weaknesses:**

- The experiments rely entirely on controlled, synthetic benchmarks. While this design enables clear isolation of factors, it is unclear how the observed mechanisms would extend to real-world settings, where additional phenomena may come into play. The degree to which these findings generalize beyond the constructed tasks is unclear.
- Several important ideas are described only verbally, without accompanying formalism. For instance, the discussion around lines 329-335 on identity-based composition equivalence could benefit from simple propositions that explicitly formalize the claims. For example, proving that identity-induced equivalence is the only possible form in the uniform benchmark; and that the two cases of zero compositional generalization (within-k without identity, and cross-k regardless of identity) follow directly from this property. Such additions would clarify the logical structure of the argument and strengthen its contribution.
- While some conclusions drawn from the experiments are well substantiated by the empirical results (especially when such behavior is predicted from theoretical analysis, such as the one described in the point above), others seem a bit handwavy/too strong to draw from the experiments. Seeing examples of failure/success in those cases would have been helpful, but still wouldn't be enough, as, for instance, in isolating the effect of module coverage, we need to know that ALL failures come from lack of module coverage. This is not clear at the moment. I might be missing something, but even if that is the case, the authors have to do a better job at discussing these sufficiency/necessity conditions, and formalizing them mathematically as possible.

Other examples of the above weakness:
- Lines 88–105: The claim that "exact and approximate equivalence in the diverse benchmark causes non-zero performance in direct models" feels overstated. Here, direct models supposedly perform better only because specific string-operation functions (e.g., mode, filter, sort) share invariances that let them superficially appear compositional. The argument attributes improved accuracy entirely to these functional properties without empirically isolating their effect. For example, it is unclear what percentage of all samples consists of examples with such invariances.
- Lines 378-385, and again in lines 462-465: The claim is that compositional equivalence encourages shortcuts in step-by-step models. Again, what percentage of the successes in predicting the final token comes from shortcuts?

**Questions:**

N/A

---

> ### Author Response · Authors · 2025-11-22
> **Response to Reviewer Mq38**
>
> We appreciate your detailed review and the time spent to provide insightful feedback.
>
> **Regarding empirically isolating the effect of composition equivalence and overstating its role:** We hope that the new results provided in Figure 6 address your concern about the empirical effect of composition equivalence. We agree that other mechanisms might be at play in a few cases and may also cause non-zero performance; e.g., in Figure 6(b), we observe that a couple of non-shared test equivalence classes have mean accuracies of 5%-6% (corresponding to the 29% shared equivalence split). We will revise lines 88-105 to more precisely characterize composition equivalence as a dominant mechanism rather than the sole driver of non-zero performance, as shown by the results in Figures 5 and 6 and the supporting evidence in Figures 14 and 15.
>
> **Module coverage and its formalization:** We hope that including a quantitative metric in Section 4 addresses your concerns about formalization and isolating the effect of module coverage. Specifically, the metric quantifies the degree to which a given test set is out-of-distribution in terms of coverage, and results in Figure 7 show that the performance can be more reliably predicted at the extreme ends of this divergence (almost 100% at low divergence values, 0% at significantly high values) and remains unreliable in between.
>
> **On necessity and sufficiency conditions for module coverage and composition equivalences and related theory:**
> In this work, we have identified two important measures of data distribution shifts—composition equivalence and module coverage—that explain substantial variability in compositional generalization performance for both direct and step-by-step models. Through extensive experiments across a wide range of train/test splits that capture varying degrees of these two kinds of shifts, we provide **strong** empirical evidence that model performance varies significantly with these shifts. We view our key contribution as empirically establishing the importance of these factors, which we consider an essential first step towards understanding compositional generalization, with a formal theoretical characterization as an important direction for future work.
>
> **Percentage of samples that exhibit shortcut learning in step-by-step models:** In Figures 4(a) and 4(b), we show the final output accuracies and step-by-step accuracies. Final accuracy denotes % of eval samples (out of 10k samples) for which final outputs are predicted correctly (regardless of intermediate outputs), and step-by-step accuracy denotes % of eval samples for which all intermediate outputs and final outputs are predicted correctly. The difference between these two accuracies corresponds to % of samples for which the model exhibits shortcut learning. For example, in Figure 4 (a) with a training permutation size of 200, the difference between step-by-step and final output accuracy is roughly 50%; hence, 50% of the samples are correctly predicted by shortcut learning. We will clarify this in the revision.
>
> **Motivation for using synthetic benchmarks:** As you also mentioned, we focused on synthetic tasks to allow systematic control over train/test splits while keeping model size smaller. As we move to more complex real-world tasks, the model size requirements also increase, making it computationally challenging to run a large number of systematic experiments, as done in this paper, to understand the role of key factors comprehensively. Another reason to focus on simple string manipulation tasks is that we wanted to focus on functions that can be represented and learned by the Transformer architecture easily (e.g., RASP primitives; Weiss et al., 2021). This design choice is inspired by a large body of work studying out-of-distribution characteristics in Transformer architecture using various mathematical and algorithmic reasoning tasks, e.g., length generalization studies (Anil et al., 2022; Zhou et al., 2024).
>
> **Implications on real-world tasks:** The sequential composition tasks considered in the paper are similar to code-output reasoning benchmarks, such as [CRUXEval](https://crux-eval.github.io/). It is observed that models such as GPT-4 can achieve up to 63% accuracy in code output reasoning. Decomposing the performance of these models due to the presence of equivalent programs during training (e.g., different implementations of the same program) is an important future direction of this work.
>
> We appreciate the other suggestions for improving clarity, and we will include them in future revisions.
>
> **References:**
> - Weiss, Gail, Yoav Goldberg, and Eran Yahav. "Thinking like transformers." International Conference on Machine Learning. PMLR, 2021.
> - Zhou, Hattie, et al. "What Algorithms can Transformers Learn? A Study in Length Generalization." The Twelfth International Conference on Learning Representations.

---

### Author Response · Authors · 2025-11-22
**Overall Response**

We thank all the reviewers for their efforts and insightful feedback. We have revised the manuscript (significant changes marked in teal; pages 8, 9, 10, and the appendix) and added new results to address the concerns raised. Below, we summarize the key changes and address each reviewer's comments through individual responses.

**New results**
1. **Isolating the effect of composition equivalence on non-zero performance of direct models for the diverse benchmark:** Both reviewers Mq38 and tN2B raise the excellent point about isolating the effect of shared composition equivalences on the performance of direct models. Following the suggestion by reviewer tN2B, we conducted a new experiment in which we systematically varied the degree of shared equivalences in the diverse benchmark and showed that composition equivalence has a significant effect on the performance of direct models.

    **Approach:** We used the following approach to create systematic splits (described on pages 8-9, l. 422-440):
      - Compute pairwise composition equivalence scores between all composition sequences.
      - Construct a connected graph from these scores and identify equivalence classes using connected components based on a threshold. We use 0.01 as a conservative measure to learn disjoint splits.
      - Create systematic train/test splits with varying degrees of shared equivalence by systematically leaking sequences from each test class into training.

    **Key findings:**
      - Figure 6(a) shows that the test-set accuracy of the direct models increases from 0% to 80% as the percentage of test sequences with shared equivalence classes increases from 0% to 100%.

      - Figure 6(b) shows the accuracy distribution within each shared/non-shared equivalence class. We see that the mean accuracy of shared equivalence classes improves as more classes are leaked into the training set. In contrast, the mean accuracy of non-shared classes remains low and constant with increasing equivalences, demonstrating a causal relationship.

      - Figure 15 (in the Appendix) provides a fine-grained visualization of two equivalence classes (out of 14) and their corresponding test accuracies, showing concrete examples of successes and failures.


2. **Quantification of module coverage:** Reviewers Mq38 and tN2B asked about formalizing module coverage. We devised a simple metric that quantifies the degree of shift between test/train and unifies the findings discussed in Section 4. Specifically:

    - **Module coverage divergence metric:** We quantify the distribution shift between test and train sets using (1) position distribution (probability that module token j appears at position i) and (2) pairwise adjacency distribution (probability that module token j is followed by j'). We measure shift using KL divergence, which captures how much modules appear at different positions or in different contexts in the test set relative to training. Module coverage divergence is defined as an equally weighted sum of position-wise and pairwise divergence.

    - **Revised Figure 7 — compositional generalization performance vs. module coverage divergence:** We computed the module coverage divergence metric across previously discussed random and systematic sampling strategies. Random sampling of 100-700 compositions (K=6) creates low coverage divergence (0.02-0.12) and near-perfect accuracy for both direct and step-by-step models. Systematic lexicographic sampling of the same number of compositions produces higher divergence (0.8-5.23), with performance degrading from perfect at low divergence to zero at high divergence.

3. **Generalization of findings to larger models:** Reviewers Mq38, ya87, and tN2B raised questions about generalizing the findings to larger models. We conducted the following experiments to answer some of those questions. (Section A.9 in the appendix)

    - **Pretrained models:** We fine-tuned and evaluated the compositional generalization performance of the pre-trained [Gemma3-1B](https://huggingface.co/google/gemma-3-1b-it) model on the following train/test splits with direct supervision on the final outputs. (Section A.9, Figures 17, 18, 19, 20)
	    - Diverse (without identity, disjoint split with 0% shared equivalences): **Test acc:** 1%
	    - Diverse (without identity, with 100% shared equivalences): **Test acc:** 93%

	    These results demonstrate that composition equivalence significantly affects generalization in larger pre-trained models as well, not just in small transformers. We focused on these extreme splits to efficiently validate the composition equivalence hypothesis, as fine-tuning across all systematic splits is computationally intensive.

    - **Larger transformer architecture (n_heads=12, n_layers=12)**: Figure 21 shows the results, and we observe similar performance trends to those seen in the smaller architecture used for most experiments (n_heads=6, n_layers=3).

---

### Meta-Review · Area_Chair_F2Ya · 2026-01-04

**Summary:**

The reviewers raised the following concerns:
- Limited benchmarks: the experiments rely entirely on controlled, synthetic benchmarks.  It is unclear how the observed mechanisms would extend to real-world settings.
- Limited emperical justification: Experiments are conducted only with small GPT-2-style models. It is unclear whether the findings generalize to larger, more capable transformers or real-world tasks.
- Lack theoretical justification: should formalizing claims mathematically; Module coverage lacks a formal theoretical definition.
- Unclear generalization to real-world settings.
- Missing ablation study to identify the source of improvement: should isolate the effect of composition equivalence; quantify module coverage, percentage of samples that exhibit shortcut learning in step-by-step models.
- Unclear contribution: The shortcut learning problem introduced in this paper is not specific to Transformers. It happens for other model architectures as well since it is more of a data problem, so the contribution is not very clear.
- Missing discussion with related work Ramesh et al. (2023)
- Missing details about training and validation data sets in the case of explaining identity-based equivalence.
- Lack in-depth discussion on the  failure results from using step supervision.

**Reviewer Concerns:**

The following reviewer concerns have been addresses partially by the rebuttal:
- Isolating the effect of composition equivalence on non-zero performance of direct models for the diverse benchmark
- Quantification of module coverage
- Generalization of findings to larger models
- Motivation for using synthetic benchmarks
- Implications on real-world tasks
- Relevance of work related to Ramesh et al. (2023)
- More details about training and validation data sets

The concerns that are still outstanding:
- Lack theoretical justification
- Limited empirical justification
- Unclear generalization to real-world settings

**Reviewer Scores:**

Reviewers Mq38, ya87 might keep their negative scores due to not fully addressed concerns.

Most of Reviewer 8EDx's concerns were addressed by the rebuttal, 8EDx might increase the score to 4.

Most of Reviewer tN2B's concerns were addressed by the rebuttal, tN2B might keep his slight positive score.

[Note] This paper exceeds 9 page limit and should be desk-rejected.

---

### Decision · Program_Chairs · 2026-01-26

Reject